# The Arctic Ocean Observation Operator for 6.9 GHz (ARC3O) - Part 2: Development and evaluation

Clara Burgard[1,2], Dirk Notz[1,3], Leif T. Pedersen[4], and Rasmus T. Tonboe[5]

[1]Max Planck Institute for Meteorology, Hamburg, Germany
[2]International Max Planck Research School for Earth System Modelling, Hamburg, Germany
[3]Institute of Oceanography, Center for Earth System Research and Sustainability, Universität Hamburg, Hamburg, Germany
[4]National Space Institute, Technical University of Denmark, Lyngby, Denmark
[5]Danish Meteorological Institute, Copenhagen, Denmark

**Correspondence:** Clara Burgard (clara.burgard@mpimet.mpg.de)

**Abstract.** The observational uncertainty in sea-ice-concentration estimates from remotely-sensed passive-microwave brightness temperatures is a challenge for reliable climate model evaluation and initialization. To address this challenge, we introduce a new tool: the Arctic Ocean Observation Operator (ARC3O). ARC3O allows us to simulate brightness temperatures at 6.9 GHz at vertical polarisation from standard output of an Earth System Model. To evaluate sources of uncertainties when
applying ARC3O, we compare brightness temperatures simulated by applying ARC3O on three assimilation runs of the MPI Earth System Model (MPI-ESM), assimilated with three different sea-ice concentration products, with brightness temperatures measured by the Advanced Microwave Scanning Radiometer Earth Observing System (AMSR-E) from space. We find that the simulated and observed brightness temperatures differ up to 10 K in the period between October and June, depending on the region and the assimilation run. We show that these discrepancies between simulated and observed brightness temperature
can be mainly attributed to the underlying observational uncertainty in sea-ice concentration and, to a lesser extent, to the data assimilation process, rather than to biases in ARC3O itself. In summer, the discrepancies between simulated and observed brightness temperatures are larger than in winter and locally reach up to 20 K. This is caused by the very large observational uncertainty in summer sea-ice concentration but also by the melt-pond parametrization in MPI-ESM, which is not necessarily realistic. ARC3O is therefore capable to realistically translate the simulated Arctic Ocean climate state into one observable
quantity for a more comprehensive climate model evaluation and initialization.

## 1  Introduction

The diversity in sea-ice concentration observational estimates affects our understanding of past and future sea-ice evolution as it inhibits reliable climate model evaluation (Notz et al., 2013) and initialization (Bunzel et al., 2016). It also limits our ability to fully exploit relationships between the evolution of sea ice and other climate variables, such as global-mean surface
temperature (Niederdrenk and Notz, 2018) and $CO_2$ emissions (Notz and Stroeve, 2016). To address these issues, we construct an observation operator for the Arctic Ocean at the frequency of 6.9 GHz. This operator provides an alternative approach for climate model evaluation and initialization with satellite observations.

Sea-ice concentration observational estimates are derived from passive microwave brightness temperature measurements from satellites. Some of the uncertainty in these estimates can be induced by the electronic noise at the level of the satellite. Most of the uncertainty is however introduced during the interpretation of the measurements because we lack simultaneous observations of the relevant physical variables having an influence on the radiation. The contribution of the individual drivers

on the brightness temperature cannot be disentangled unambiguously if the properties describing the state of the ice, snow, open ocean surface, and atmosphere are not available simultaneously. Nevertheless, a variety of algorithms have been developed to retrieve an estimate of sea-ice concentration from the measured brightness temperatures. These retrieval algorithms take advantage of the fact that the relative influence of the individual physical variables on the brightness temperature depends on the frequency and polarization of the radiation. Using different combinations of measurements at various frequencies and

polarizations, the retrieval algorithms result in a range of sea-ice concentration products, which differ however, sometimes substantially (Ivanova et al., 2014; Kern et al., 2019). The evaluation of simulated sea-ice concentration in General Circulation Models (GCMs) is therefore influenced by the choice of the sea-ice concentration product against which a simulation is evaluated (Notz et al., 2013).

Observation operators applied to general circulation model (GCM) output have been suggested as a solution to circumvent

this observational uncertainty for other climate variables (Flato et al., 2013; Eyring et al., 2019). An observation operator enables us to simulate brightness temperatures based on output from a GCM. This simulated brightness temperature can then be evaluated against the observed brightness temperature. Uncertainty in the evaluation can therefore only be induced by uncertainties in the observation operator and remaining electronic noise. We argue that the simulated brightness temperature based on a GCM is a more consistent method less prone to uncertainty than the use of retrieval algorithms because the GCM

provides an internally consistent climate state over time and space. Additionally, the simulation of the brightness temperature relies on several physical variables. Therefore, the evaluation of the simulated brightness temperature is an evaluation of the physical climate state as a combination of several variables, allowing an integrated evaluation of the simulated climate state of the GCM instead of a "variable-to-variable" evaluation.

However, the simulation of sea-ice brightness temperatures relies on sea-ice properties not explicitly resolved in most GCMs.

In particular, brightness temperatures are driven by the vertical brine distribution inside the ice and liquid water distribution in the snow, which are driven by temperature and salinity profiles. However, Burgard et al. (2020) showed in a one-dimensional idealized setup that, using a few simple assumptions, the low complexity of GCM output is sufficient to simulate reasonable sea-ice brightness temperatures at 6.9 GHz at vertical polarization. While we focus on the frequency of 6.9 GHz in this study, we suggest that a similar methodology to the framework proposed here, together with the framework presented in Burgard

et al. (2020), can be used to investigate the simulation of brightness temperatures at other frequencies in the future as well. However, the increasing influence of snow on the brightness temperature with increasing frequency and the limited possibilities of simulating snow properties in a GCM remain a challenge to overcome first.

In this study, we first present an Arctic Ocean observation operator that we construct based on the suggestions from Burgard et al. (2020). We then evaluate the brightness temperatures simulated based on assimilation runs against brightness temperatures

observed by satellites and investigate potential uncertainty sources in the brightness temperature simulation.

## 2   The Max Planck Institute Earth System Model

As a baseline for the development of an Arctic Ocean observation operator, we use the Max Planck Institute Earth System Model (MPI-ESM). It is a state-of-the-art Earth System Model that contributed to the Coupled Model Intercomparison Project in its fifth phase (Taylor et al., 2012) and will contribute to its sixth phase (Eyring et al., 2016). We use its low resolution configuration (MPI-ESM-LR).

The atmosphere component, ECHAM6 (Stevens et al., 2013), has a horizontal resolution of T63 (~1.9° x 1.9°) and a vertical division into 47 levels between surface and 0.01 hPa. The ocean component, MPIOM (Jungclaus et al., 2013), is based on a curvilinear grid with two poles located in South Greenland and Antarctica. The horizontal resolution ranges from 15 km near Greenland to 185 km in the tropical Pacific. Vertically, the ocean is divided into 40 levels between surface and bottom. The sea ice is simulated within MPIOM by a dynamic/thermodynamic sea-ice model based on Hibler (1979). In this simple setup, the sea-ice salinity is kept constant at 5 g/kg, and the ice bottom temperature is kept constant at -1.8 °C. There is no explicit simulation of the ice thickness distribution. Still, the simulation of the mean state and variability of Arctic sea ice is realistic (Notz et al., 2013).

For our observation operator, we use output from the atmosphere component ECHAM6. The sea-ice properties, such as sea-ice concentration, sea-ice thickness and snow thickness, are computed within the ocean component and communicated to the atmosphere component through coupling on a daily frequency (Jungclaus et al., 2013). Based on these properties, ECHAM6 computes the snow cover fraction and the melt pond coverage (Giorgetta et al., 2013), which are needed for a comprehensive assessment of the radiative properties of the surface. Additionally, ECHAM6 provides the atmospheric water and ice content, which are needed for the calculation of the radiation path through the atmosphere (see Sec. 3.2). ECHAM6 therefore provides all variables needed for the simulation of Arctic Ocean brightness temperatures.

## 3   The Arctic Ocean Observation Operator ARC3O

The purpose of the ARCtic Ocean Observation Operator for 6.9 GHz (ARC3O) is to simulate Arctic Ocean brightness temperatures as could theoretically be measured at the top of the atmosphere of a climate model by a satellite rotating around that climate model. This brightness temperature is a result of radiation emitted by the surface, upwelling atmospheric radiation, reflected downwelling atmospheric radiation, atmospheric transmission, and reflected space radiation (Swift and Cavalieri, 1985).

As a consequence, ARC3O is based on two parts. In the first part, an emission model computes the sea-ice surface brightness temperature (see Sec. 3.1). In the second part, an atmospheric radiative transfer model combines the sea-ice surface emission with ocean emission and atmospheric emission, reflection and transmission (see Sec. 3.2). The workflow of ARC3O follows five steps (see Fig. 1), which we explain in the following.

**ARC3O workflow**

1    Prepare masks for season and ice types

*GCM*
Sea-ice thickness
Snow thickness
Surface temperature → Processing for each point →

*Mask for seasons*
Melting snow, bare summer ice, cold conditions

*Mask for ice types*
First-year ice, multiyear ice

2    Prepare sea-ice profiles for cold conditions

*GCM*
Sea-ice thickness
Snow thickness
Surface temperature

*Masks (ice types and seasons)* → Processing for cold conditions points →

*Snow-covered ice profiles*
Layer temperature, salinity, thickness, wetness, density, correlation length, snow/first-year/multiyear ice

*Bare ice profiles*
Layer temperature, salinity, thickness, wetness, density, correlation length, first-year/multiyear ice

3    Compute sea-ice surface brightness temperature for cold conditions

*Snow-covered ice profiles* → MEMLS → *Snow-covered ice brightness temperature* 

*Bare ice profiles* → MEMLS → *Bare ice brightness temperature*

x snow-cover fraction / x bare fraction → *Cold conditions ice brightness temperature*

4    Compute sea-ice surface brightness temperature for all conditions

*Cold conditions ice brightness temperature*

*Masks (seasons)* → Processing for each point →

*Ice brightness temperature*
*Cold conditions:* Cold conditions ice brightness temperatures
*Melting snow:* Snow surface temperature
*Summer bare ice:* Constant inferred from observations (266.78 K)

5    Add sea-ice concentration and atmospheric effect

*GCM*
Sea-ice concentration
Melt pond fraction
Snow-ice column surface temperature
Sea surface temperature
Atmospheric columnar liquid water and water vapor

*Ice brightness temperature* → Simple ocean emission and atmospheric radiative transfer model (Wentz and Meissner , 2000) → *Brightness temperature at top of atmosphere*

**Figure 1.** Workflow of the Arctic Ocean Observation Operator ARC3O.

## 3.1 The contribution of the sea-ice surface to the brightness temperature

The brightness temperature $TB_{ice}$ emitted at an ocean surface completely covered by sea ice at 6.9 GHz, vertical polarization, is primarily driven by the vertical distribution of the brine volume fraction inside the ice, which principally depends on the temperature and salinity profile (see Burgard et al. (2020) for more details). However, MPI-ESM does not provide temperature and salinity profiles. To circumvent this lack of information, we follow the suggestion of Burgard et al. (2020) to build a simple model for these sea-ice properties, based on the boundary conditions given by the GCM.

### 3.1.1 Identifying different periods and ice types

Burgard et al. (2020) showed that the simulation of sea-ice surface brightness temperatures relies on different assumptions, depending on the conditions of the ice. A sea-ice year can be divided into three periods: cold conditions, melting snow, and bare ice near 0 °C. Additionally, sea-ice brightness temperatures depend on the ice type, for example first-year or multiyear ice.

We therefore flag the different type of periods and different ice types based on the sea-ice properties given by the MPI-ESM output (Step 1 in Fig 1). Grid cells containing melting snow are flagged as "melting snow periods", grid cells containing bare ice in July, August and September are flagged as "bare ice near 0°C", and the remaining grid cells are flagged as "cold conditions". To flag the different grid cells as "first-year ice", "multiyear ice" and "open water only", we consider the ice thickness evolution. If the ice thickness is zero, the ice type is set to "open water only", if the ice thickness is larger than zero but there has been at least one "open water only" timestep in the year preceding the timestep evaluated, the ice type is set to "first-year ice". If none of the two before apply, the ice type is set to "multiyear ice". This is a simplification, assuming that there is no ice drift and that the ice present at one point in time and space will be the same, but older, ice later in time.

### 3.1.2 Cold conditions

In periods of cold conditions, Burgard et al. (2020) showed that the sea-ice surface brightness temperature can be simulated using an emission model by assuming a linear vertical temperature profile and a function of depth for the salinity. They also showed that the uncertainty range does not vary substantially if the profiles are interpolated to five, seven or ten layers. In this study, we choose to use ten layers. We construct profiles (Step 2 in Fig 1), divided into eleven layers, namely ten layers of ice and one layer of snow. The ice layers are equidistant, based on the ice thickness given by MPI-ESM, and the snow layer thickness is equal to the snow thickness given by MPI-ESM.

We construct temperature profiles based on the ice surface temperature given by MPI-ESM, which represents the temperature at the top of the snow and ice column. For each grid cell, we construct two sets of profiles. One set of profiles interprets the surface temperature as the snow surface temperature. This profile is a combination of two linear profiles, one in the snow, defined by the snow thermal conductivity, and one in the ice, defined by the ice thermal conductivity, based on the temperature

$T_{\text{ice,surf}}$ at the interface between ice and snow inferred as follows:

$$T_{\text{ice,surf}} = \frac{T_{\text{snow,surf}} \cdot \frac{k_s}{h_s} + T_{\text{bottom}} \cdot \frac{k_i}{h_i}}{\frac{k_s}{h_s} + \frac{k_i}{h_i}} \tag{1}$$

with $k_s$ the thermal conductivity of snow (= 0.31 W/Km), $k_i$ the thermal conductivity of ice (= 2.17 W/Km), $h_s$ the snow thickness, $h_i$ the ice thickness, $T_{\text{snow,surf}}$ the temperature at the surface of the snow, $T_{\text{bottom}}$ the temperature at the bottom of the ice, set to -1.8 °C.

The other set of profiles interprets the surface temperature as the ice surface temperature and is a linear profile between surface and bottom temperature. The ice bottom temperature is taken as constant at -1.8 °C in both cases.

The salinity profile $S$ is computed as a function of depth $z$, as formulated by Griewank and Notz (2015), based on the results of 1D simulations with the complex thermodynamic sea-ice model SAMSIM and their comparison to observations. $S$ is defined as follows for first-year ice:

$$S_{fy}(z) = \frac{z}{a + bz} + c \tag{2}$$

with a = 1.0964, b = -1.0552 and c = 4.41272

and as follows for multiyear ice:

$$S_{my}(z) = \frac{z}{a} + \left(\frac{z}{b}\right)^{1/c} \tag{3}$$

with a = 0.17083, b = 0.92762 and c = 0.024516.

We set the snow salinity to zero. Note, however, that the validity of this assumption is slightly uncertain as the lowest layer of the snow can be saline, especially above first-year ice (Barber et al., 1998; Shokr and Sinha, 2015; Nandan et al., 2017), enabling the presence of brine at the base of the snow.

The vertical profile of the ice density $\rho_i$ is computed based on the vertical temperature $T$ and salinity $S$ profiles, with the following formula applied to each ice layer (Notz, 2005):

$$\rho_i = \Phi_b \cdot \rho_w + (1 - \Phi_b) \cdot \rho_0 \tag{4}$$

where $\Phi_b$ is the brine volume fraction:

$$\Phi_b = \begin{cases} S/S_b & \text{if } S_b > 0 \text{ - Eq. (1.5) in Notz (2005)} \\ 1 & \text{if } S_b = 0 \end{cases} \tag{5}$$

$S_b$ is the brine salinity:

$$S_b = \begin{cases} 508.18 + 14.535T + 0.2018T^2 & \text{if } T \in [-43.2°\text{C}, -36.8°\text{C}] \text{ - Eq. (39) in Vant et al. (1978)} \\ 242.94 + 1.5299T + 0.04529T^2 & \text{if } T \in [-36.8°\text{C}, -22.9°\text{C}] \text{ - Eq. (39) in Vant et al. (1978)} \\ -1.20 - 21.8T - 0.919T^2 & \text{if } T \in ]-22.9°\text{C}, -8.0°\text{C}[ \text{ - Eq. (3.4) in Notz (2005)} \\ 1/(0.001 - (0.05411/T)) & \text{if } T \in [-8.0°\text{ C}, 0°\text{C}[ \text{ - Eq. (3.5) in Notz (2005)} \\ 0 & \text{if } T = 0 \end{cases} \tag{6}$$

$\rho_w$ is the density of brine with the chemical composition of seawater at 0 °C (Eq. (3.8) in Notz, 2005):

$$\rho_w = 1000.3 + 0.78237 S_b + 2.8008 \cdot 10^{-4} S_b^2 \tag{7}$$

$\rho_0$ is the density of pure ice (Pounder, 1965):

$$\rho_0 = 916.18 - 0.1403 T \tag{8}$$

The density of the snow layer is set to 300 kg/m$^3$, like in MPI-ESM (Giorgetta et al., 2013).

The vertical profile of the correlation length, a measure for the scatterer size (snow particles, brine inclusions, air bubbles), depends on the ice type. If the ice layer is in the upper 20 cm of first-year ice, the correlation length is set to 0.35 mm. If the ice layer is located below the upper 20 cm, the correlation length is set to 0.25 mm (Tonboe, 2010). For multi-year ice, the correlation length is set to 1.5 mm for all ice layers (Burgard et al., 2020). The correlation length of the snow layer is set to 0.15 mm (Tonboe, 2010).

The sea-ice surface brightness temperature is simulated based on the temperature, salinity, density, thickness, and correlation length profiles described above. A slightly modified version of the Microwave Emission Model for Layered Snowpacks (MEMLS, Wiesmann and Mätzler, 1999) extended for sea ice (Tonboe et al., 2006) is used for the brightness temperature simulation. It relates the snow and ice properties to emission, absorption and scattering of the microwave radiation in each layer. Hence, MEMLS simulates the path of the radiation through the ice and snow from bottom to top, resulting in a brightness temperature emitted at the surface.

MPI-ESM provides a snow cover fraction, which means that the ice is not always fully covered by snow. To account for the effect of both snow-covered ice and bare ice on the radiation, we simulate two sea-ice surface brightness temperatures for each grid cell. One set of brightness temperatures is simulated using the temperature profiles computed through snow and ice, where the surface temperature given by MPI-ESM is interpreted as the snow surface temperature. The other set is simulated using linear temperature profiles computed through ice only, assuming that there is no snow cover on the ice and the surface temperature given by MPI-ESM is interpreted as the ice surface temperature (Step 3 in Fig 1). These surface brightness temperatures are then combined, weighted by the snow cover fraction given by MPI-ESM, resulting in one mean sea-ice surface brightness temperature.

### 3.1.3 Melting snow

In spring, temperatures increase across the Arctic Ocean, leading to the melting of the snow covering the sea ice. Wet snow strongly affects the emitted microwave radiation. This effect mainly depends on the water content and on the density of the snow (Chang and Gloersen, 1975; Ulaby et al., 1986; Shokr and Sinha, 2015). However, these snow properties are not resolved in MPI-ESM. We therefore cannot use MEMLS to simulate the brightness temperature of the ice and snow column in this case. Instead, we use the following definition of the brightness temperature:

$$\text{TB} = \epsilon_{\text{eff}} \cdot T_{\text{eff}} \tag{9}$$

where $\epsilon_{\mathrm{eff}}$ is the emissivity of the emitting part of the ice and snow column, i.e. the layers influencing the resulting radiation emitted at the surface, and $T_{\mathrm{eff}}$ the integrated temperature over this same emitting part (Hallikainen and Winebrenner, 1992; Tonboe, 2010; Shokr and Sinha, 2015). At 6.9 GHz, the emitting part of wet snow is a thin subsurface layer and the emissivity is close to 1 (Hallikainen et al., 1986; Lee et al., 2018). Following Eq. 9, we therefore assume that the brightness temperature
5    of ice covered by melting snow is equal to the snow surface temperature.

### 3.1.4    Summer bare ice near 0 °C

In summer, after the snow has fully melted away, the salinity profile inside the ice cannot necessarily be represented by a simple function of depth. As the subsurface ice can be assumed to be isothermal close to 0 °C during summer, the brine volume fraction in the subsurface layer, i.e. in the upper few cm of the ice, increases and melt ponds form. Above a subsurface
10   brine volume fraction of 0.2, the brightness temperature is proportional to the subsurface brine volume fraction (Burgard et al., 2020). The subsurface brine volume fractions above 0.2 can be interpreted as a measure for the melt-pond fraction as they mean that the surface is very wet. For these warm conditions, the physical properties of the ice which is not covered by melt ponds are similar over the whole Arctic Ocean. Burgard et al. (2020) therefore suggest a very simple approach: use a constant brightness temperature for the ice surface fraction, i.e. the ice fraction not covered by melt ponds.

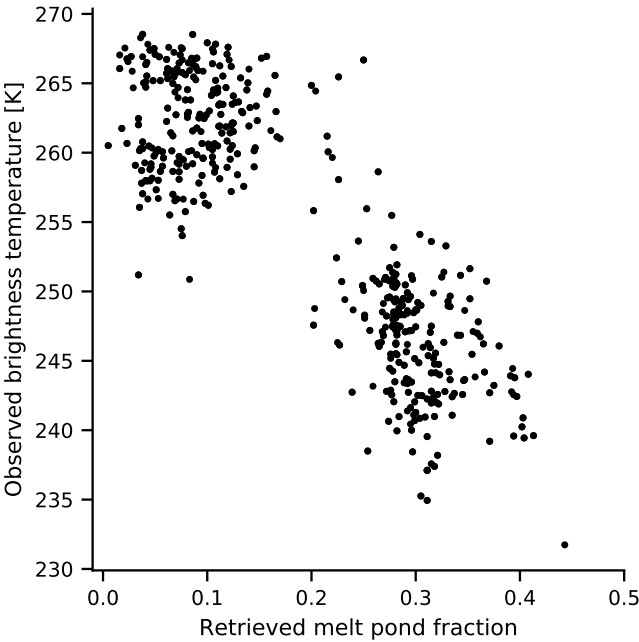

**Figure 2.** Brightness temperatures measured by AMSR2 against the melt pond fraction product by Istomina et al. (2015b) for different points represented in the Round Robin Data Package from May to mid-August 2011.

To find a surface brightness temperature representing the sea-ice surface in summer, we use the observational dataset Round Robin Data Package (RRDP, Pedersen et al., 2018) developed as part of the European Space Agency (ESA) sea-ice Climate Change Initiative (SICCI). These data cover the period from May to mid-August 2011. The RRDP contains amongst others microwave brightness temperatures between 6 and 89 GHz measured by the Advanced Microwave Scanning Radiometer 2

(AMSR2) collocated with the melt-pond fraction product by Istomina et al. (2015b) over areas estimated to be close to 100% sea-ice concentration. Using the combination of the melt-pond fraction and the brightness temperatures measured from space, we can infer the summer brightness temperature of melt-pond-free sea ice (Fig. 2). Burgard et al. (2020) showed that, above a brine volume fraction of 0.2 in the upper few cm of ice, the brightness temperature is proportional to this subsurface brine volume fraction. We assume that melt ponds have the same effect as high subsurface brine volume fractions. Below 0.2, the

effect of the ice on the brightness temperature is stronger than the effect of melt ponds. Therefore, to infer an ice brightness temperature for non-ponded ice, we take the mean brightness temperature of ice covered by 20% or less melt pond. This method results in a summer ice brightness temperature of 262.29±3.56 K at the top of the atmosphere. After applying an atmospheric correction (see Sec. 3.2), the resulting summer ice brightness temperature at the surface is 266.78 K.

## 3.2 The contribution of ocean and atmosphere to the brightness temperature

As the Arctic Ocean is not covered by 100% of sea ice, the brightness temperature measured at the top of the atmosphere is also influenced by the relative fraction and properties of open water surfaces and properties of the atmosphere. To take into account these oceanic and atmospheric contributions, we use a geophysical model developed by Wentz and Meissner (2000) (Step 5 in Fig. 1).

In this geophysical model, the total brightness temperature of an Arctic Ocean grid cell is computed as a combination

of the upwelling surface emission by ocean, sea ice, and melt ponds, the upwelling atmospheric emission, the atmospheric transmittance, the atmospheric emission reflected by the different types of surface, and the reflected background radiation from space. The ocean surface brightness temperature is computed as a function of surface temperature, surface salinity, and wind speed. The melt-pond brightness temperature is computed similarly to the ocean brightness temperature, but setting salinity and wind speed to zero. Finally, as the atmosphere is mostly transparent to radiation in the low microwave range, the radiative

transfer through the atmosphere is computed based only on the columnar water vapor and columnar cloud liquid water.

The sea-ice surface brightness temperature used as input for this geophysical model is set for each grid cell depending on the three periods presented above (Step 4 in Fig. 1). The sea-ice surface brightness temperature is computed by MEMLS (see Sec. 3.1.2) in cold conditions and approximated with the snow surface temperature in conditions of melting snow (see Sec. 3.1.3). In summer, however, our method was based on brightness temperatures as measured by satellites from space and

therefore resulted in a sea-ice brightness temperature for the top of the atmosphere. To obtain a sea-ice brightness temperature at the surface that can be used as input for the combination of ocean and sea-ice surface brightness temperature for summer conditions, we need to apply an atmospheric correction.

To infer a mean atmospheric correction, we apply the geophysical model to regions covered by 99% or more sea ice in MPI-ESM output presented in Sec. 4.2, setting all melt pond fractions to zero. This way, we have no influence by open water

surfaces, be it ocean or melt ponds, on the resulting brightness temperature. We set the ice surface brightness temperature used as input for the geophysical model to a random constant. We then subtract this constant ice surface brightness temperature from the top-of-the-atmosphere brightness temperature simulated by the geophysical model based on atmospheric properties given by the climate model output. This gives us a mean atmospheric effect of 4.49 K. We add this to the brightness temperature of

262.29 K inferred in Sec. 3.1.4, resulting in a constant brightness temperature of 266.78 K as a constant brightness temperature representing the radiation emitted at the summer bare ice surface. This is the bare ice summer surface brightness temperature that can be used for combination with open water (ocean and melt ponds) brightness temperature in the geophysical model in Step 5 of Fig. 1.

## 4    Evaluation of ARC3O

The approach we use to construct ARC3O was proposed by Burgard et al. (2020), based on an idealized one-dimensional setup that did not involve actual observations. In the following, we evaluate our simulated Arctic Ocean brightness temperatures against brightness temperatures measured by satellites.

We do so by comparing brightness temperatures simulated by ARC3O based on MPI-ESM output from assimilation exper-iments, i.e. experiments where the model is regularly nudged towards observations. Hence, we expect the simulated climate

system to be close to reality and the simulated brightness temperature to be close to the observed brightness temperature.

However, the observations used in the data assimilation are reanalysis data for the atmosphere and ocean and retrieved sea-ice concentration products for the sea ice. They are therefore not direct observations but already-processed products prone to differences to reality. Additionally, in the assimilation process, MPI-ESM is nudged towards observations but some character-istic features inherent to the mean model state might remain. This is the case when the mean model state and the assimilated

state are so incompatible that the model will rapidly drift back towards the mean model state. The uncertainty of the measured brightness temperature itself is around 1 K (NASDA, 2003) and thus neglected here. Hence, differences between observed and simulated brightness temperature can arise from three sources: (1) the difference between real and retrieved climate state due to the difference between retrieval algorithms or reanalysis and the real climate state, (2) the difference between the assimilated climate state and the retrieved or reanalysis product, and (3) biases in ARC3O (Fig. 3). In the following, we try to quantify

how the first two uncertainty sources contribute to differences between the simulated and observed brightness temperatures. Any remaining biases can then be attributed to biases of ARC3O itself.

### 4.1    Observation data

As observed brightness temperatures, we use Calibrated Passive Microwave Daily EASE-Grid 2.0 (CETB) brightness temper-atures processed as part of the NASA Making Earth System Data Records for Use in Research Environments (MEaSUREs)

program (Brodzik et al., 2016, Updated 2018). They are an improved, enhanced-resolution, gridded passive microwave Earth System Data Record (ESDR) for monitoring cryospheric and hydrologic time series from the measurement devices Scanning Multi-channel Microwave Radiometer (SMMR), Special Sensor Microwave Imager/Sounder (SSM/I-SSMIS) and Advanced

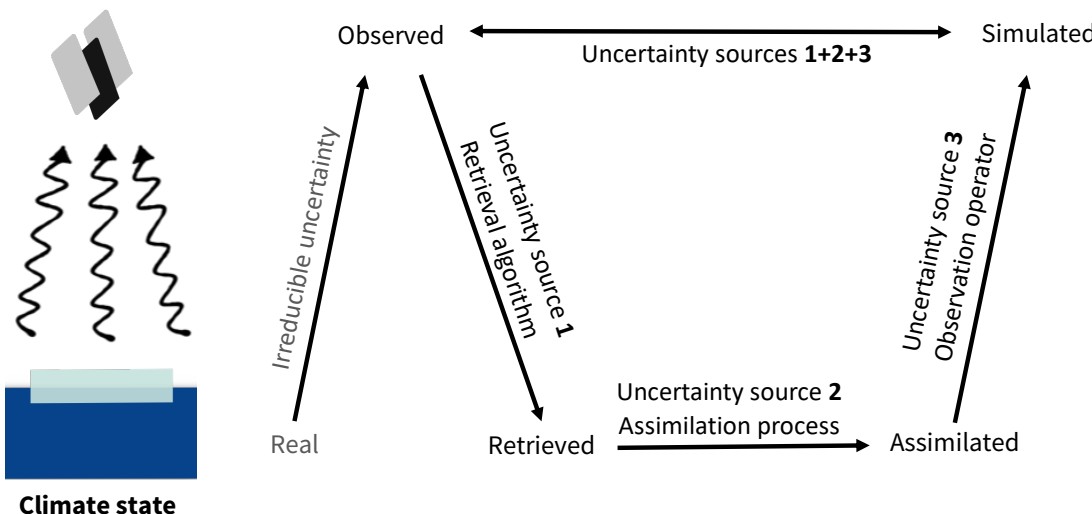

**Figure 3.** Uncertainty sources possibly introducing differences between simulated and observed brightness temperature.

Microwave Scanning Radiometer - Earth Observing System (AMSR-E). These data cover the period between 1978 and 2017 and are provided on a 25 km x 25 km grid. For the comparison with MPI-ESM data, we focus on the period from 2002 to 2008 and interpolate the observations bilinearly to the model grid (1.9°x1.9°). Again, we concentrate on the frequency of 6.9 GHz, vertical polarization. At this frequency and this time period, the observations stem from AMSR-E.

## 4.2 Model data

We use model data from assimilation runs, as they are nudged towards the observed climate state and are therefore expected to be a reasonable estimate of the real climate state in the model. Differences between simulated and observed brightness temperatures should therefore be small and can be attributed to the three uncertainty sources presented before (see Fig. 3). To examine the impact of the choice in sea-ice retrieval product, we use three assimilation runs based on three different sea-ice concentration products. The atmosphere and the ocean component are assimilated in the same way in all three cases.

The assimilation experiments cover the period from 2002 to 2008 and were conducted by Bunzel et al. (2016). The assimilation technique used was Newtonian relaxation, also called nudging. Atmospheric, oceanic and sea-ice properties were nudged into the model using full-field data assimilation in all atmospheric and oceanic levels. In the atmosphere, vorticity, divergence, temperature, and surface pressure were nudged into the model with a relaxation time of one day, while salinity and temperature in the ocean were nudged with a relaxation time of ten days. For the assimilation of atmospheric quantities, the ERA-Interim dataset (Dee et al., 2011) was used, while the ocean was nudged toward Ocean Reanalysis System 4 data (Balmaseda et al., 2013).

For sea ice, only sea-ice concentration was assimilated. The three different sea-ice concentration products are the ESA SICCI Version 2 (SICCI2) dataset (Lavergne et al., 2019) as a 50-km-gridded product, and the NASA Team dataset (Cavalieri et al., 1996) and the Bootstrap dataset (Comiso, 2000), both as 25-km-gridded products. We choose these datasets because SICCI2 is a new algorithm combining several existing algorithms with the goal of improving the retrieved sea-ice concen-

tration product, Bootstrap sea-ice concentrations are in the upper range of retrieved sea-ice concentrations, and NASA Team sea-ice concentrations are in the lower range (Ivanova et al., 2014; Kern et al., 2019). The data were interpolated bilinearly to the model grid before assimilation. In grid boxes containing missing values, e.g. the polar observation hole (northward of 87.2°N), no assimilation was applied. The sea ice was then exclusively calculated by the model. To avoid brightness temperature uncertainties due to this free simulation region, we mask out the region northward of 86.72°N, the highest latitude on our

grid below the observational hole. The relaxation time was 20 days. Relaxation times differ among the model components to account for the different response times of the components. In order to allow for a realistic relation between ice concentration $SIC$ and thickness $h$, sea-ice thickness was updated in the model proportionally to ice concentration nudging (Tietsche et al., 2013). The assimilation changes the thickness $h$ in the given grid cell by $\Delta h_{\mathrm{assim}}$, which is proportional to $\Delta \mathrm{SIC}_{\mathrm{assim}}$, with a proportionality factor $h*$ of 2 m, as follows:

$$\Delta h_{\mathrm{assim}} = h * \Delta \mathrm{SIC}_{\mathrm{assim}} \qquad\qquad (10)$$

### 4.3 Cold seasons (JFM, AMJ, OND)

#### 4.3.1 Comparison between simulated and observed brightness temperatures

The first comparison between simulated and observed brightness temperatures clearly showed a positive bias over the whole Arctic Ocean in the simulated brightness temperatures (Fig. A1, left). The brightness temperature is defined as the product of

the emissivity and the physical temperature of the emitting part of the ice (Ulaby et al., 1986). A comparison of the simulated emissivities with emissivities derived from observational data from the RRDP showed that ARC3O systematically overestimates the emissivity. It is however not straightforward to find where the bias is produced in the emission model. We therefore chose to correct the bias by multiplying the inherent sea-ice emissivity by a tuning coefficient at the end of step 3 of the ARC3O workflow (see Fig. 1). The coefficient which yields the best agreement with observations is 0.968 (Fig. A1, right).

More information about the tuning process is found in App. A. In the following, we discuss brightness temperatures simulated with this tuning procedure.

The three different sets of simulated brightness temperatures show largely similar behaviours in the cold seasons winter (January/February/March, JFM), spring (April/May/June, AMJ), and autumn (October/November/December, OND) (Fig. 4). Overall, differences between simulated and observed brightness temperatures are very small and are generally lower than 10 K.

The pattern of differences appears to be similar across seasons. The simulated brightness temperatures are slightly higher than the observed ones in regions of high sea-ice concentration and thickness, e.g. north of the Canadian Archipelago and the Central Arctic in winter. In contrast, they are lower than the observed ones in regions of low sea-ice concentration and thickness, e.g. in the marginal zones such as the Barents Sea, the Pacific sector, and the Hudson Bay in the brightness temperature simulations

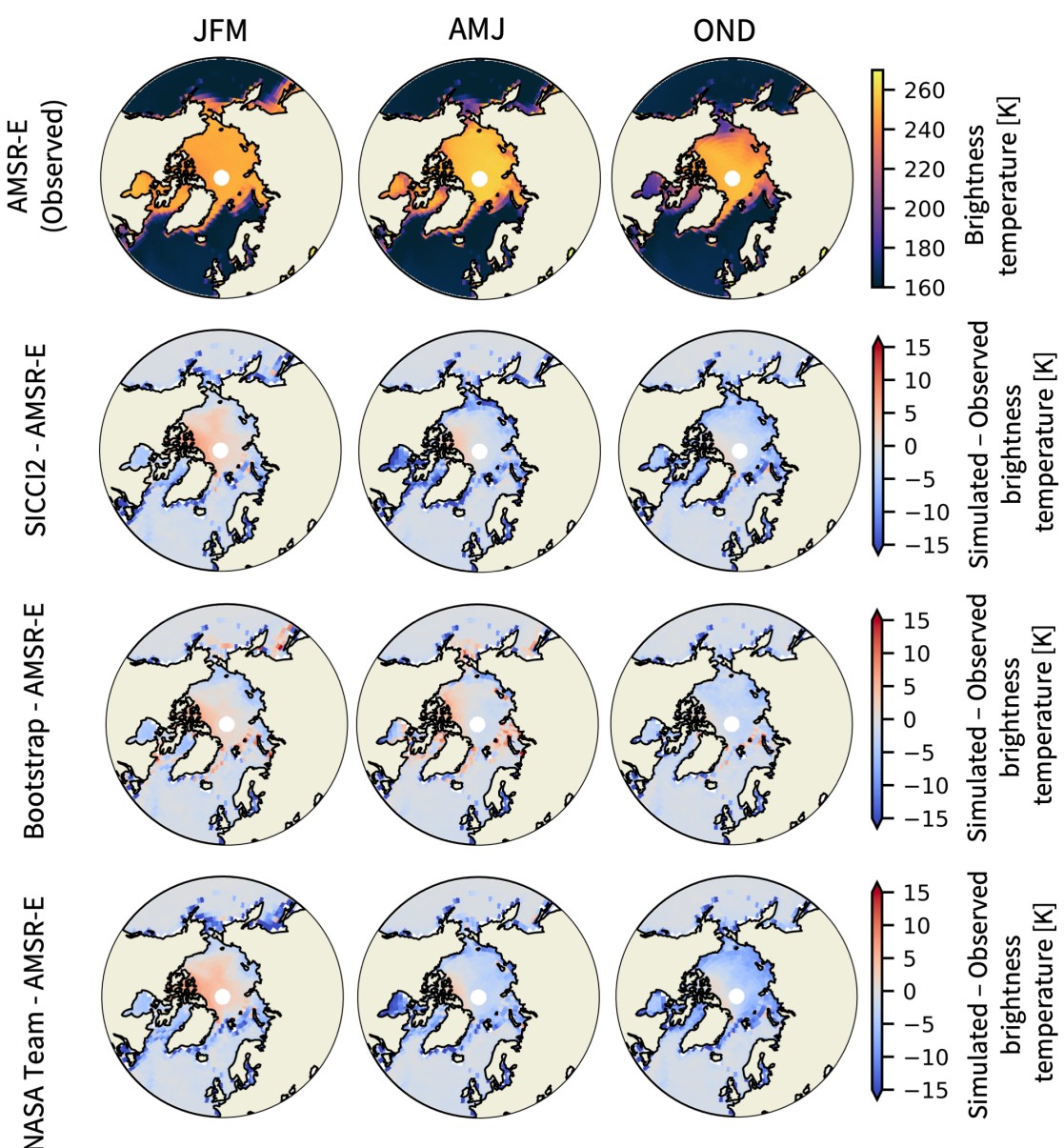

**Figure 4.** Observed brightness temperatures by AMSR-E (1st row). Differences between brightness temperatures simulated with ARC3O from MPI-ESM output assimilated with SICCI2 (2nd row), Bootstrap (3rd row) and NASA Team (4th row) sea-ice concentration and observed brightness temperatures. The columns stand for the three cold seasons: JFM, AMJ, OND. Summer (JAS) is discussed in Sec. 4.4.

based on the SICCI2 and NASA Team assimilation runs, and higher in the brightness temperature simulation based on the Bootstrap assimilation run.

The overestimation on the order of 2 to 4 K in the Central Arctic in winter has a similar pattern in all three sets of simulated brightness temperatures. Otherwise, brightness temperatures simulated based on the Bootstrap assimilation run are very close to the observed ones, with differences to the observations of usually less than 3 K. Only a few individual points in the Atlantic sector show larger biases. Brightness temperatures simulated based on the NASA Team and SICCI2 assimilation run show

stronger differences to observations. The simulated brightness temperatures are up to 10 K lower than the observations in the North Pacific in winter and up to 15 K lower than the observations in the Hudson Bay in spring. In the Central Arctic and the Atlantic Sector, the brightness temperatures simulated based on the NASA Team assimilation run are 2 to 5 K lower than observations in spring and 5 to 10 K lower than observations in autumn. The pattern of differences between brightness temperatures simulated based on SICCI2 assimilation run and observations is similar to the pattern of differences between

brightness temperatures simulated based on the NASA Team assimilation run and observations but the brightness temperatures simulated based on the SICCI2 assimilation run are about 2 K higher than the brightness temperatures simulated based on the NASA Team assimilation run.

### 4.3.2 Investigating uncertainty sources

The total difference $\Delta_{\text{tot}}$ between simulated and observed brightness temperatures is a consequence of the difference between

real and retrieved climate state $\Delta_{\text{retriev}}$, of the difference between retrieved and simulated climate state $\Delta_{\text{assim}}$, and of biases in the brightness temperature simulation by ARC3O $\Delta_{\text{ARC3O}}$ (Fig. 3):

$$\Delta_{\text{tot}} = \Delta_{\text{retriev}} + \Delta_{\text{assim}} + \Delta_{\text{ARC3O}} \tag{11}$$

We set out to investigate $\Delta_{\text{assim}}$ and $\Delta_{\text{retriev}}$ to gain an estimate of $\Delta_{\text{ARC3O}}$.

In a first step, we investigate which drivers the brightness temperature is particularly sensitive to. We concentrate on the

variables provided by MPI-ESM, as these are the ones we can quantify in our setup. In the cold seasons, the most important drivers for the simulation of a brightness temperature for a surface covered by varying fractions of sea ice and open water are the sea-ice concentration, sea-ice thickness, snow thickness, and surface temperature.

We examine the sensitivity for the month of October, representing the beginning of the freezing period, and for the month of March, representing the end of the freezing period. We do so for the SICCI2, the Bootstrap, and the NASA Team assimilation

run. As the results of the analysis are similar for all three assimilation runs unless otherwise mentioned, we only show the results for the SICCI2 assimilation run in the following.

We start by estimating the internal variability for each variable at a given date and at a given grid cell. For each date in October and March, we compute the standard deviation over a sample of five years (2003 to 2008), for each variable and each grid cell. As an overview, the range and the time mean of the resulting variabilities are shown in Table 1 (first column) and in

Fig. B1, respectively.

With these variability fields, we conduct sensitivity studies for each variable of interest. For both October and March, we conduct two sets of experiments per variable of interest, one in which we add the variability field to the variable of the reference assimilation run and one in which we subtract the variability field. Ranges of the difference between the resulting brightness

**Table 1.** Sensitivity of the simulated brightness temperature to different input variables of the simulation. We show the spatial 5th and 95th percentile of the time mean of the estimated variability for the variables of interest (also see Fig. B1) in the 1st column. We show the spatial 5th and 95th percentile of the time mean difference between modulated and reference brightness temperature when the variability field is added to (2nd column) and subtracted from (3rd column) the reference field. These values are inferred from the SICCI2 assimilation run.

| | Range of variability estimates | Range of variation in brightness temperature | |
| --- | --- | --- | --- |
| | | Increase in variable | Decrease in variable |
| **October** | | | |
| Sea-ice concentration | ±0 to 28 % | 0 to 16.9 K | -19.9 to 0 K |
| Sea-ice thickness | ±2 to 43 cm | -0.4 to 0 K | -0.2 to 0.4 K |
| Snow thickness | ±0 to 6 cm | 0 to 0.4 K | -0.6 to 0 K |
| Surface temperature | ±0.4 to 5.3 K | 0.09 to 1.6 K | -1.7 to -0.1 K |
| **March** | | | |
| Sea-ice concentration | ±0 to 25 % | 0.3 to 18.8 K | -17.0 to -0.2 K |
| Sea-ice thickness | ±3 to 41 cm | -0.9 to 0 K | -0.1 to 1.0 K |
| Snow thickness | ±0 to 11 cm | 0 to 1.0 K | -1.4 to 0 K |
| Surface temperature | ±0.5 to 7.1 K | 0.03 to 2.8 K | -3.0 to -0.04 K |

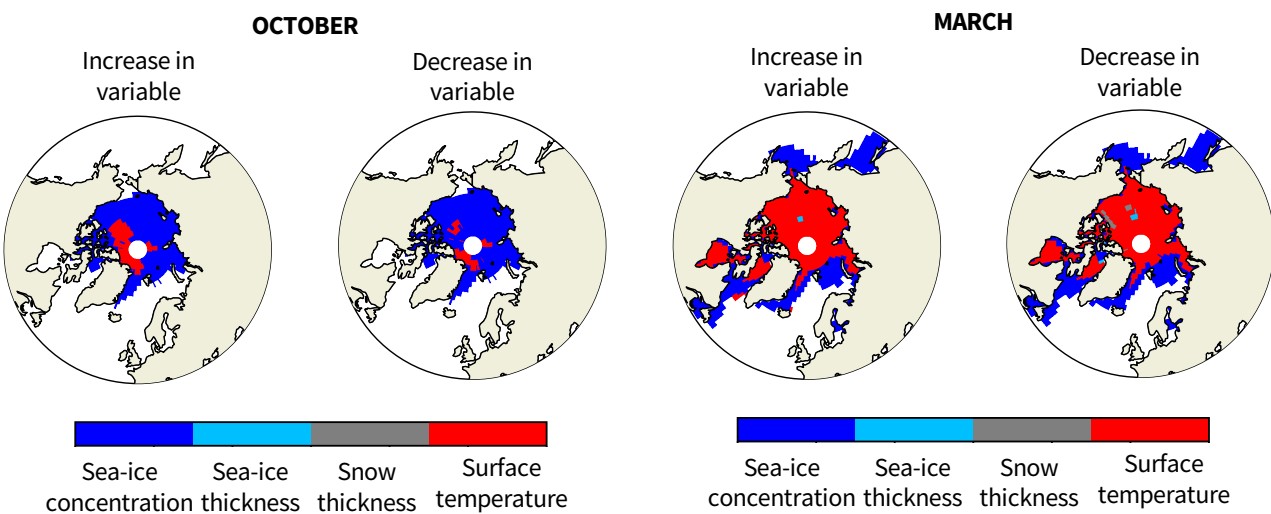

**Figure 5.** Variable which has the highest absolute mean effect on the brightness temperature in October (left) and March (right) when their variability field (see Table 1 and Fig. B1) is added to (1st and 3rd column) and subtracted from (2nd and 4th column) the input variable. These results are inferred from the SICCI2 assimilation run. Inspired from Fig. 5 in Richter et al. (2018).

temperatures and the brightness temperatures simulated based on the reference assimilation run are shown in Table 1 (2nd and 3rd column). The main message emerging from the results is that the sea-ice concentration variability is the driver for the largest variations in the brightness temperature, changing it by up to nearly 20 K, while the variability in other variables affects the brightness temperature only up to 3 K. Spatially, the sea-ice concentration is the main driver for variability in regions

not completely covered by ice (Fig. 5). In regions covered by near to 100% of ice, the surface temperature has the highest effect on the brightness temperature. Although they both have an indirect influence on the ice surface temperature as well, sea-ice thickness and snow thickness do not play, except very locally, an important role for uncertainties in the total brightness temperature of a grid cell in the simulated variability range as their mean absolute contribution to the brightness temperature variability is on the order of 1 K.

In the sensitivity study using the NASA Team assimilation run, the variability range of sea-ice thickness and surface temperature are comparable to the other runs (not shown). However, the sea-ice thickness is the main driver of variability for a large region in the Central Arctic north of Alaska in March (not shown). Further investigation into these differences in the main driver for the brightness temperature could lead to a better understanding of the differences in the simulated climate of these three assimilation runs. However, this is beyond the scope of our study and is a subject for future work.

In summary, this sensitivity study shows that sea-ice concentration and surface temperature are the dominant drivers of variability in the simulated brightness temperature. To understand the total uncertainty $\Delta_{\text{tot}}$, we therefore need to focus on these two variables.

In a next step, we investigate the influence of $\Delta_{\text{assim}}$ on $\Delta_{\text{tot}}$. The goal of a data assimilation is to reach a simulated climate state close to reality. During the data assimilation process, the model is nudged towards three distinct observational datasets:

an ocean reanalysis, an atmosphere reanalysis, and a sea-ice concentration product, which are not necessarily consistent with each other. Hence, discrepancies can arise between the variables before and after the assimilation. This is the case for example when a non-zero sea-ice concentration is assimilated at one point but the ocean temperature is too warm to sustain the ice at that point and the ice directly melts away.

As the sea-ice concentration is the main driver for uncertainties in the brightness temperature simulation, we here focus on

the effect of the data assimilation procedure on the sea-ice concentration in the three different assimilation runs. This effect is mostly visible in the marginal regions (Fig. 6) and is of similar magnitude for all three assimilation runs. At the ice edge, the differences between the original sea-ice concentration observational product and the sea-ice concentration assimilated in the simulation are highest, on the order of 5 %. As a rule of thumb, differences of 1% in sea-ice concentration are equivalent to differences of 1 K in brightness temperatures (see Burgard et al., 2020), so the differences in sea-ice concentration are roughly

equivalent to resulting differences in brightness temperature of around 5 K. $\Delta_{\text{assim}}$ can therefore account for a large part of the total difference between simulated and observed brightness temperature $\Delta_{\text{tot}}$ in the ice edge region (see Fig. 4).

Unfortunately, the difference between real and retrieved sea-ice concentration $\Delta_{\text{retriev}}$ cannot be as robustly quantified as $\Delta_{\text{assim}}$. In-situ observations for a robust evaluation of the retrieved sea-ice concentration products are largely lacking. Although there have been local evaluation approaches, based on regions known to be covered by 100% sea ice (e.g. Ivanova et al., 2015;

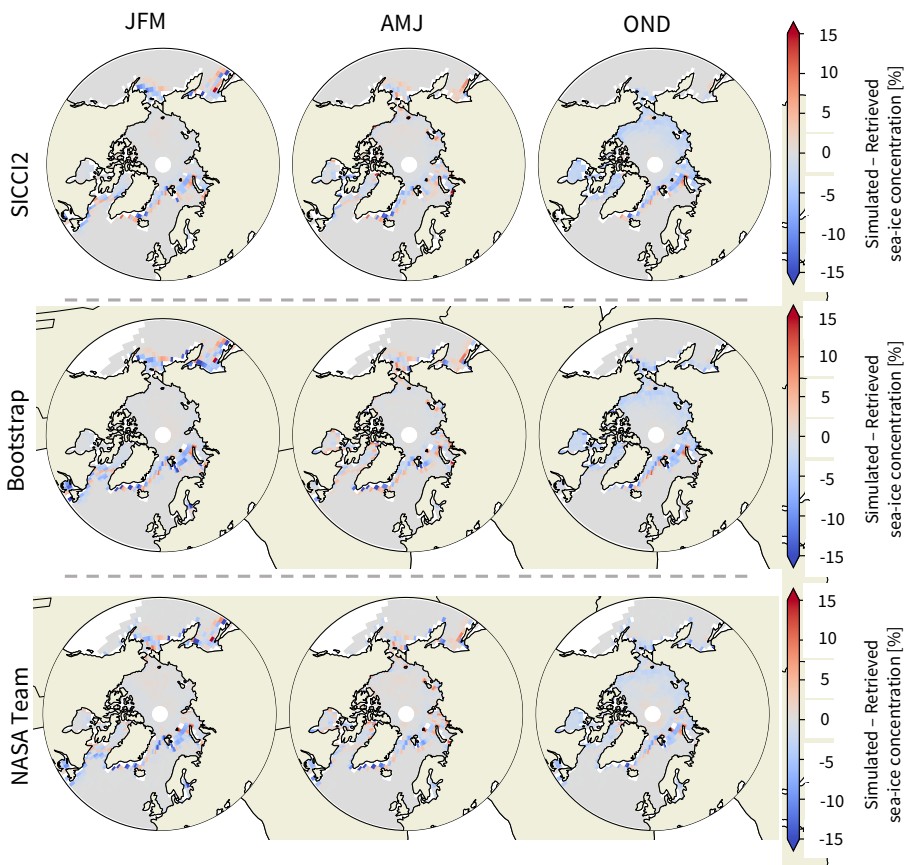

**Figure 6.** Difference between retrieved (before data assimilation) and simulated (after data assimilation) sea-ice concentration for the three assimilation runs.

Tonboe et al., 2016; Lavergne et al., 2019), this lack of in-situ observations inhibits the evaluation of the products on larger scales against reality. The effect of $\Delta_{\text{retriev}}$ and $\Delta_{\text{ARC3O}}$ can therefore not clearly be disentangled.

Still, we can give an estimated range for $\Delta_{\text{ARC3O}}$. To do so, we use the spread between the SICCI2, Bootstrap and NASA Team sea-ice concentrations as an estimate of the uncertainty range around the real sea-ice concentration (Fig. 7 and Ivanova et al., 2014; Kern et al., 2019). First, we subtract $\Delta_{\text{assim}}$ from the simulated brightness temperature. $\Delta_{\text{tot}}$ is now only a sum of $\Delta_{\text{retriev}}$ and $\Delta_{\text{ARC3O}}$. Second, for each grid cell and each time step, we evaluate if the observed brightness temperature is located within the range of the brightness temperatures simulated based on the three different sea-ice concentration products. If yes, differences are not necessarily a bias induced by ARC3O. If not, it is likely that ARC3O induces a bias. In this case, the simulated brightness temperature with the lowest absolute distance from the observed brightness temperature represents the smallest plausible estimate of $\Delta_{\text{ARC3O}}$ (Fig. 8, 2nd row). The largest absolute difference between simulated and observed brightness temperatures in contrast gives an estimate of the largest plausible value of $\Delta_{\text{ARC3O}}$ (Fig. 8, 3rd row). If the observed

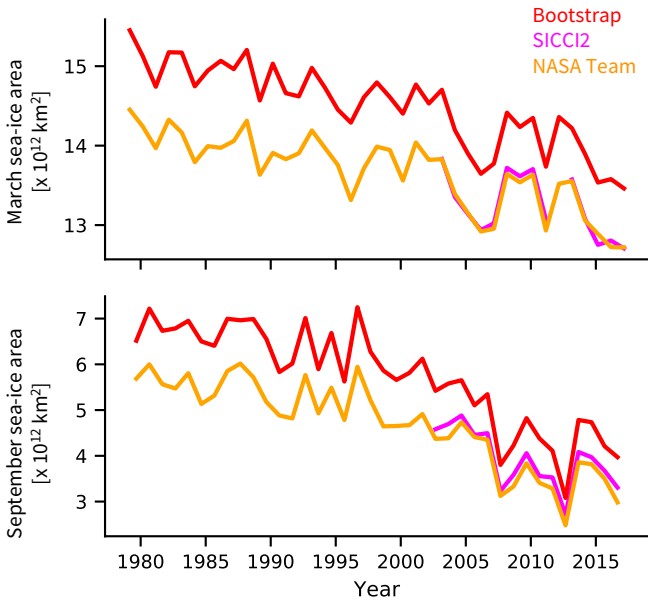

**Figure 7.** March (top) and September (bottom) sea-ice area for the three observational datasets assimilated in the three assimilation runs used.

brightness temperature is within the range obtained for the different retrieved sea-ice concentration estimates, the estimate of $\Delta_{\mathrm{ARC3O}}$ is set to zero.

The resulting mean estimates of $\Delta_{\mathrm{ARC3O}}$ are both very small, as the minimal estimates are well below 5 K and the maximal estimates are 5 K or below, except in the Hudson Bay. Additionally, the comparison of the $\Delta_{\mathrm{ARC3O}}$ estimates to the spread in sea-ice concentration between Bootstrap and NASA Team (Fig. 8, 4th row) shows that biases in ARC3O, i.e. $\Delta_{\mathrm{ARC3O}}$, are small compared to the uncertainty in retrievals, i.e. $\Delta_{\mathrm{retriev}}$, except in the Central Arctic on the Canadian side.

The overestimation on the Canadian side of the Central Arctic is likely due to a MPI-ESM bias in the sea-ice thickness. The sea-ice thickness in our assimilation runs is on the order of 2 m at its thickest north of the Canadian Archipelago (not shown). Observational estimates tend to show thicknesses of rather 3 to 4 m in this region (Ricker et al., 2017). This is much more than we varied in our sensitivity study and points to a possible stronger influence of sea-ice thickness on the difference between simulated and observed brightness temperature than estimated from the sensitivity study in this case.

The remaining uncertainty contained in $\Delta_{\mathrm{ARC3O}}$ can have several sources: ARC3O itself, further biases in the simulated climate state, or wrong assumptions in our approach. Biases in ARC3O itself can arise from wrong assumptions in the sea-ice emission model MEMLS or in the ocean emission and atmospheric transfer model by Wentz and Meissner (2000). For example, the brightness temperatures simulated over open ocean by ARC3O tend to be around 3 K too low. More investigation and fine-tuning above the open ocean might therefore slightly lower the uncertainty introduced by ARC3O. Another example is the approach to define first-year and multiyear ice in the ARC3O framework. The definition we use does not take into account

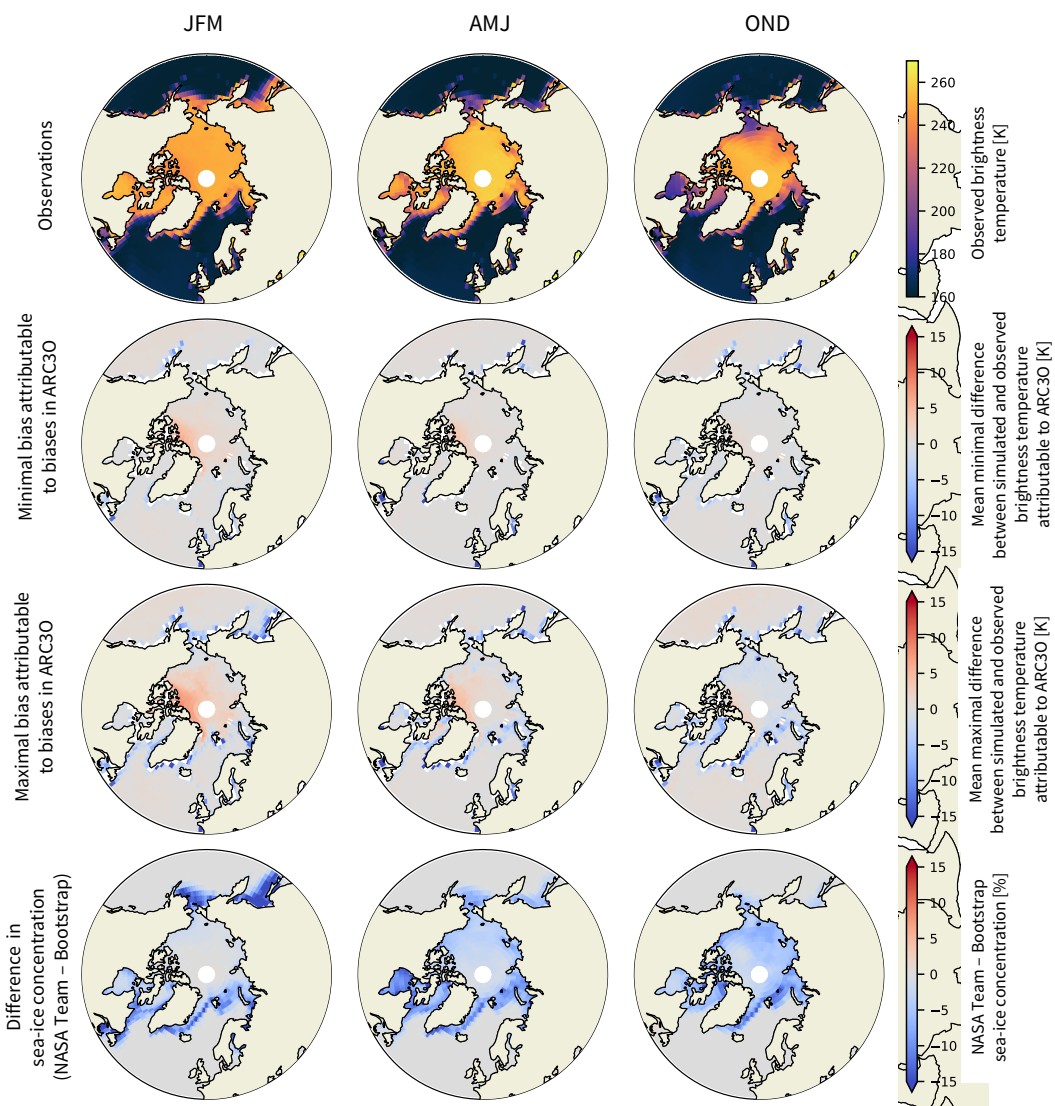

**Figure 8.** Observed brightness temperatures (1st row), mean minimal (2nd row) and mean maximal (3rd row) estimates of $\Delta_{\mathrm{ARC3O}}$, and differences in sea-ice concentration (4th row) between the NASA Team and Bootstrap assimilation runs, i.e. maximal estimates of $\Delta_{\mathrm{retriev}}$.

the dynamics of the ice. As a consequence, if a grid cell is located in a region where sea-ice circulates horizontally and this grid cell therefore contains ice for more than a year, the ice in this grid cell will be defined as multiyear ice. This is the case even if the ice transported through the grid cell is not the same physical ice floe throughout this time period but a different first-year ice floe every day for example. Rethinking this definition, e.g. by finding a way to follow the movement of the ice in the climate model, might therefore also reduce the uncertainty.

Concerning the simulated climate state, we expect small biases for the simulated surface temperature, because the ERA-Interim reanalysis compares well to the few available in-situ observations (Lindsay et al., 2014) and satellite-retrieved temperature measurements could be used if further evaluation was needed. We currently cannot explore the additional impact of model biases in sea-ice thickness and snow thickness as in-situ observations are rare and satellite retrievals of these variables are not necessarily robust yet. Observational estimates for sea-ice and snow thickness are mainly based on retrieval algorithms, similar to sea-ice concentration estimates. Possible biases may therefore remain in these variables compared to reality.

Finally, we treat the spread between the SICCI2, Bootstrap and NASA Team sea-ice concentration as an estimate of the uncertainty range around the real sea-ice concentration. Hence, we implicitly assumed that the real sea-ice concentration lies in the range between the SICCI2, the Bootstrap and the NASA Team estimates. As only limited evaluation against reality is possible, the real sea-ice concentration could lie outside of this range and the uncertainty between real and retrieved sea-ice concentration might be different to what we assume here.

As a conclusion, we showed in the consistent model setup that the sea-ice concentration is the main driver for large variations in the brightness temperature in regions that are not fully ice-covered. In regions where the sea-ice concentration is very high and does not vary much, such as the Central Arctic in winter, the surface temperature is the main driver of variations in the brightness temperature. Simulated and observed brightness temperatures are generally in good agreement. Most differences are likely driven by the uncertainty brought by the sea-ice concentration products compared to reality. Remaining differences attributable to biases in ARC3O remain below 5 K.

The lack of evaluation possibilities for the observation operator is an indicator for how little is actually known about the real Arctic climate state and processes at work, in particular in regard to the real sea-ice concentration and surface temperature. Extending this observation operator to lower and higher frequencies would be of advantage to fill this gap. Brightness temperatures at different frequencies and polarizations are sensitive to different particular parameters. The combination of different brightness temperatures could enable a comprehensive assessment of the Arctic Ocean surface and atmosphere, and a comprehensive evaluation of the individual observation operators. For example, using an observation operator applied to reanalysis data, Richter et al. (2018) simulated brightness temperatures at the frequency of 1.4 GHz. They found that sea-ice concentration and surface temperature are the main drivers for variations in the brightness temperature in the Central Arctic but that, in regions of thin ice, the ice thickness is the dominating driver at this frequency. Combining frequencies in this case would then enable a climate model evaluation encompassing different perspectives.

## 4.4 Melting season (JAS)

As in winter, the simulated summer brightness temperatures are primarily a function of sea-ice concentration. Melt ponds are then the main challenge for sea-ice retrieval algorithms, as their passive microwave signature is undistinguishable from open water. This leads to large uncertainties and potential underestimation of the sea-ice concentration in summer (Cavalieri et al., 1990; Comiso and Kwok, 1996; Fetterer and Untersteiner, 1998; Meier and Notz, 2010; Rösel et al., 2012b; Kern et al., 2016). The difference between observed and simulated brightness temperature (Fig. 9, 2nd row) and the difference between

observational sea-ice concentration products is therefore much larger in summer than in winter (Ivanova et al., 2015; Kern et al., 2016).

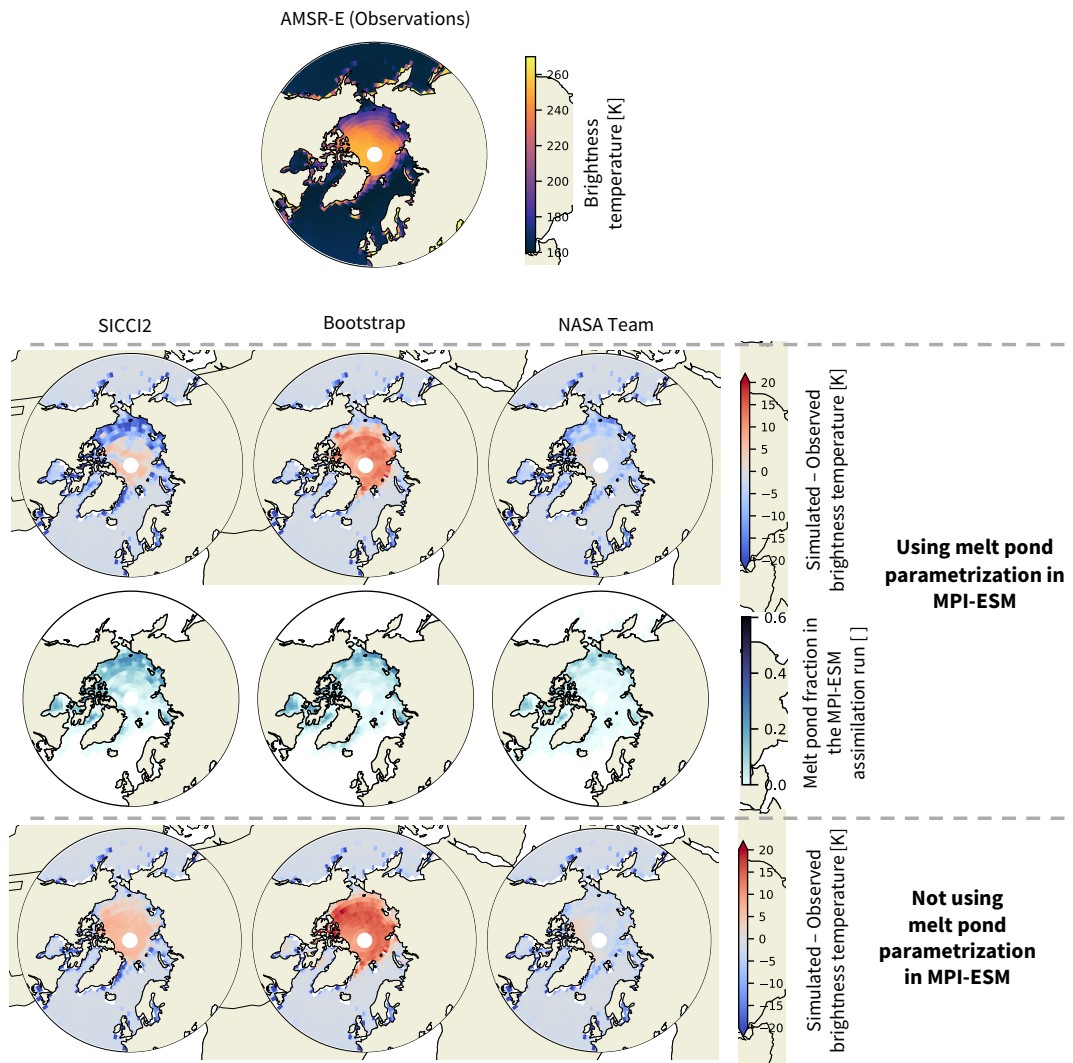

**Figure 9.** Experiment to compare brightness temperatures in summer (July/August/September) simulated based on assimilation runs assimilated with different sea-ice concentration products (SICCI2, Bootstrap, NASA Team) to brightness temperatures measured by AMSR-E. Observations by AMSR-E are shown in the 1st row. Difference between simulated and observed brightness temperatures if melt ponds are taken into account (2nd row) and if they are set to zero (4th row). In the 3rd row, the melt pond fraction simulated by MPI-ESM is shown.

In summer, the simulation of brightness temperatures in ARC3O is only based on the combination of a constant bare ice brightness temperature and melt-pond brightness temperature, weighted by the melt-pond fraction. The constant bare ice brightness temperature of 266.78 K is derived by using observed brightness temperatures and melt-pond fractions, as explained in

Sec. 3.1.4). Assumptions about the chosen constant ice surface brightness temperature can influence the analysis but we assume that the uncertainty between simulated and observed brightness temperatures in summer is mainly driven by two parameters: the difference between real and retrieved sea-ice concentration and the difference between real and simulated melt-pond fraction.

In the following, we can therefore evaluate the sea-ice concentration products in summer and their relationship to the melt pond fraction. To do so, we distinguish between two types of sea-ice concentration: the total sea-ice concentration and the pond-free sea-ice concentration. The pond-free sea-ice concentration is the concentration of sea ice visible by the satellite, assuming that melt ponds are open water. In MPI-ESM, we know both the total sea-ice concentration and the pond-free sea-ice concentration as melt ponds are represented through a melt-pond parametrization, which is a function of the surface

energy budget and water drainage to the ocean (Roeckner et al., 2012). In the SICCI2 algorithm, melt ponds are not explicitly accounted for but the dynamic tie-points are based on observed brightness temperatures in areas of high sea-ice concentration, which are covered by melt ponds in summer. The retrieved sea-ice concentration will therefore implicitly account for the melt-ponds, potentially reducing the underestimation of the sea-ice concentration (Kern et al., 2016; Lavergne et al., 2019). In the Bootstrap algorithm, a correction is applied to the sea-ice concentration to account for the effect of melt ponds by

synthetically increasing the retrieved sea-ice concentration (Comiso and Kwok, 1996; Bunzel et al., 2016), while in the NASA Team algorithm, no correction is applied (Comiso et al., 1997; Bunzel et al., 2016). By switching on and off the melt-pond parametrization in MPI-ESM, we can evaluate the ability of observational products to produce a reasonable pond-free and total sea-ice concentration.

    In this experiment, we run ARC3O on the three MPI-ESM assimilation runs setting the melt-pond fraction to zero every-

where. We then compare this set of simulated brightness temperatures to observed brightness temperatures (Fig. 9, 4th row) and to the set of brightness temperatures simulated taking into account the melt-pond distribution simulated by MPI-ESM (Fig. 9, 2nd row). The results give different insights depending on the sea-ice concentration product used for the assimilation.

    Using the SICCI2 assimilation run, the simulated brightness temperature of the pond-free sea ice is higher than the observed brightness temperature in the Central Arctic. Melt ponds cover the whole ice-covered Arctic Ocean (Rösel et al., 2012a;

Istomina et al., 2015a). Adding their effect in the brightness temperature simulation could therefore reduce the difference between simulated and observed brightness temperature. However, while the melt-pond parametrization in MPI-ESM reduces the overall brightness temperature, the reduction is very heterogeneous so that the brightness temperature is now largely underestimated in the Pacific sector but still overestimated in most of the Central Arctic. This means that the dynamic tie-point approach of the SICCI2 algorithm seems to take into account the effect of melt ponds in a reasonable way, therefore yielding a

too high pond-free sea-ice concentration. The brightness temperatures simulated using the melt-pond parametrization suggest that SICCI2 does not represent well the total sea-ice concentration. However, the melt-pond distribution in MPI-ESM seems to be too heterogeneous and therefore unrealistic in some regions, as most simulated melt ponds concentrate in the Pacific sector and not many can be found over the Central Arctic (see Fig. 9, 3rd row, and Roeckner et al., 2012). The latitudinal gradient is realistic, showing more melt ponds in lower latitudes, where the incoming solar radiation and air temperatures are higher, than

in higher latitudes. However, the melt pond fraction in the Central Arctic was observed to be slightly higher than simulated

and the melt pond fraction in some of the marginal regions was observed to be slightly lower than simulated (Rösel et al., 2012a; Istomina et al., 2015a). Corrections to the simulated melt pond fraction in these directions might reduce the simulated brightness temperature in the Central Arctic and increase it in marginal regions, approaching the observed brightness temperature. As a consequence, it seems that the total sea-ice concentration might be well represented in the SICCI2 dataset but we cannot robustly confirm this assumption with our setup due to the apparently somewhat unrealistic melt-pond parametrization provided by MPI-ESM.

Using the Bootstrap assimilation run, if the ice is assumed to be pond-free, the simulated brightness temperature is more than 10 K higher than the observed brightness temperature over the whole Central Arctic. Due to this large difference, adding melt ponds on top of the ice is not sufficient to counteract this overestimation of the brightness temperature, which remains on the order of 10 K. This means that Bootstrap tends to overestimate both the pond-free and total sea-ice concentration in summer.

Using the NASA Team assimilation run, the simulated brightness temperature of the pond-free ice is very close to the observed brightness temperature. As a consequence, the addition of melt ponds leads mainly to a negative bias compared to observations. This means that the NASA Team dataset represents well the pond-free sea-ice concentration, in agreement with previous results by Ivanova et al. (2015) but that it tends to underestimate the total sea-ice concentration in summer.

The main conclusions show that the main driver for differences between simulated and observed summer brightness temperatures are again the differences between retrieved and real sea-ice concentration. However, the melt-pond parametrization used in MPI-ESM is too heterogeneous and unrealistic and therefore contributes to the difference between simulated and observed brightness temperatures as well. For further analysis, the melt-pond parametrization could however be replaced by a climatology using observational melt-pond estimates, such as Rösel et al. (2012a) or Istomina et al. (2015a). This could reduce the uncertainty induced by the melt-pond parametrization.

## 5 Conclusions

In this study, we present the first observational operator for the Arctic Ocean that is applied to GCM output, following suggestions from Burgard et al. (2020). It allows us to simulate brightness temperatures at a frequency of 6.9 GHz, vertical polarization, for the whole Arctic Ocean. The results look promising and open up possibilities for further and deeper analysis of simulated and observed Arctic climate state.

In cold seasons, the simulation of the ice surface brightness temperature relies on the sea-ice and snow emission model MEMLS. In periods of melting snow, the emissivity of the snow surface is assumed to be close to 1, so the ice surface brightness temperature is assumed to be equal to the snow surface temperature. In summer, a constant bare ice brightness temperature is weighted with the melt-pond fraction. All these ice surface brightness temperatures are then used as input for an ocean emission and atmospheric radiative transfer model to result in a brightness temperature as could theoretically be measured at the top of the atmosphere by a satellite flying around the model.

Simulated and observed brightness temperatures compare well. In winter, differences between observed and simulated brightness temperatures attributable to biases in ARC3O are well below 5 K. In comparison, the total difference between observed and simulated brightness temperatures ranges from well below 5 K up to 10 K. The large differences can be attributed to possible differences between real and retrieved climate state, especially in sea-ice concentration, and, to a lower extent, to the process of data assimilation into the model. In summer, the difference between simulated and observed brightness temperatures locally reach more than 15 K. This difference can be attributed to high differences in the underlying sea-ice concentration products and potential biases arising from the melt-pond parametrization in the climate model.

The low estimate of the uncertainty induced by the observation operator ARC3O itself in the comparison between simulated and observed brightness temperatures shows that it is possible to simulate realistic brightness temperatures based on simple output of a GCM. This is a necessary step to open the way for similar observation operators for different frequencies and polarizations and, as a consequence, for new climate model evaluation and model initialization techniques in a hindcast or unconstrained model run. Additionally, ARC3O can be used to evaluate observation products against satellite measurements by using assimilation runs.

## 6 Outlook

An observation operator translates a consistent climate state into one observable quantity. In climate model evaluation, the full simulated Arctic climate state can therefore be evaluated against one observed quantity instead of several different retrieved quantities, which all carry uncertainties with them, especially in the Arctic region (Jakobson et al., 2012; Lindsay et al., 2014; Ivanova et al., 2015; Boisvert et al., 2018). With one observation operator at one single frequency, not all effects can be disentangled clearly, e.g. in this case the influence of sea-ice concentration and surface temperature in the Central Arctic are comparable. Further development of observational operators for different frequencies is essential to use this approach to its fullest. A multi-frequency framework would allow us to investigate this consistent climate state from different perspectives, as different variables affect different frequencies differently. Also at 6.9 GHz, further investigation and work might improve the brightness temperature simulation.

The possibility of representing the climate state of a climate model in only one observable quantity is also very beneficial to model initialization through data assimilation. The first-guess procedure used in data assimilation methods, such as variational data assimilation (Talagrand and Courtier, 1987; Andersson et al., 1994) or ensemble Kalman filters (Evensen, 1994; Hunt et al., 2007), would then be based on a consistent climate state and be conducted in observation space, independent of retrieval algorithms (Scott et al., 2012; Richter et al., 2018). This is already done and has led to improvements in weather prediction systems for other regions than the Arctic (e.g. Terasaki and Miyoshi, 2017).

The observational uncertainty of the sea-ice concentration is very large in summer. We showed here that, if we are able to reduce the uncertainty in the melt-pond representation of the model, we can relate differences between observed and simulated brightness temperatures directly to differences between retrieved and real total sea-ice concentration. This is a promising perspective as melt ponds are a strong challenge for the retrieval of summer sea-ice concentrations.

Also, in this study, we investigated the Arctic Ocean climate as simulated by a GCM with a simple sea-ice model. A few other GCMs use more detailed sea-ice modules (Vancoppenolle et al., 2009; Bailey et al., 2018), including e.g. an ice thickness distribution within a single grid cell. Using ARC3O in combination with these more detailed sea-ice modules could give further insights into the importance of small-scale thickness variations on the brightness temperature.

5    Finally, ARC3O is a simple observation operator as it is based on variables simulated by all GCMs, can be applied to output from any kind of GCM simulation, and does not require extensive computational power. It is therefore a powerful tool which has the potential to uncover model biases and improve model initialization by providing a new perspective on the Arctic climate system.

*Code and data availability.* Primary data and scripts used in this study are archived by the Max Planck Institute for Meteorology and can be
10   obtained by contacting publications@mpimet.mpg.de. The code of ARC3O can be downloaded under https://github.com/ClimateClara/arc3o.

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

## Appendix A: Tuning of the temperature profiles

The brightness temperatures initially produced by ARC3O were clearly too bright (Fig. A1, left). A comparison of the simulated emissivities with emissivities derived from observational data from the RRDP showed that ARC3O systematically overestimates the emissivity. The brightness temperature is defined as the product of the emissivity and the physical temperature of the emitting part of the ice (Ulaby et al., 1986). As it is not straightforward to find where the bias is produced in the emission model, we chose to multiply the inherent sea-ice emissivity with a tuning coefficient to counteract the systematic bias. To do so, we selected all points with a sea-ice concentration of 99.7% or more to avoid influence from open water, in the year 2004. We then multiplied the sea-ice surface brightness temperature by a range of coefficients between 0.96 and 0.975. We found the best agreement between simulated and observed brightness temperatures for a coefficient of 0.963 in the months January, February, and March and for a coefficient of 0.973 in the months October, November, December. As a consensus, we therefore chose a coefficient of 0.968 to apply to the sea-ice brightness temperatures, which yields a more reasonable distribution of brightness temperatures (Fig. A1, right).

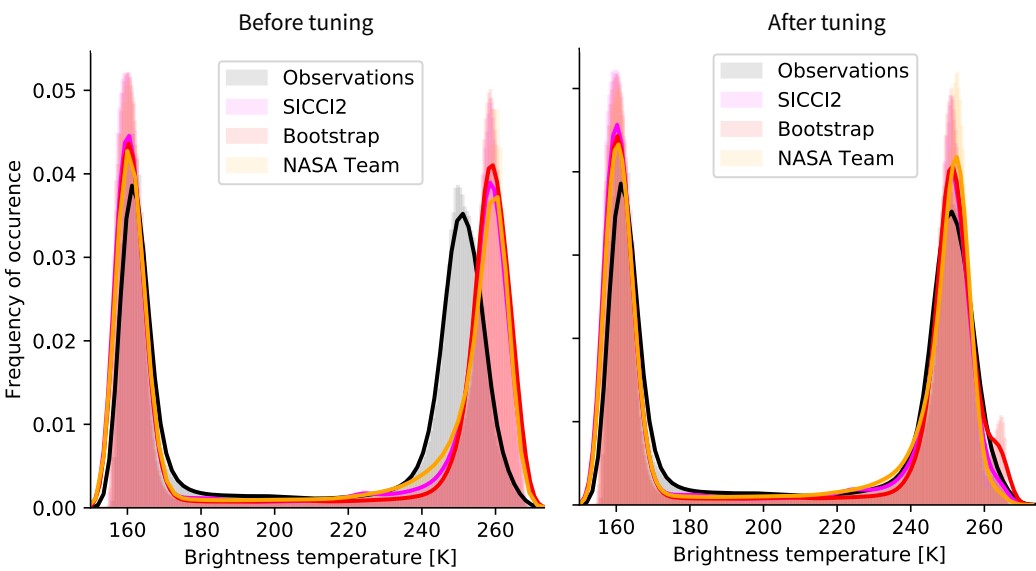

**Figure A1.** Density distribution of the brightness temperatures in the three simulated cases and in observations in the untuned (left) and tuned (right) version for the years 2005 to 2008.

## Appendix B: Variability in climate parameters

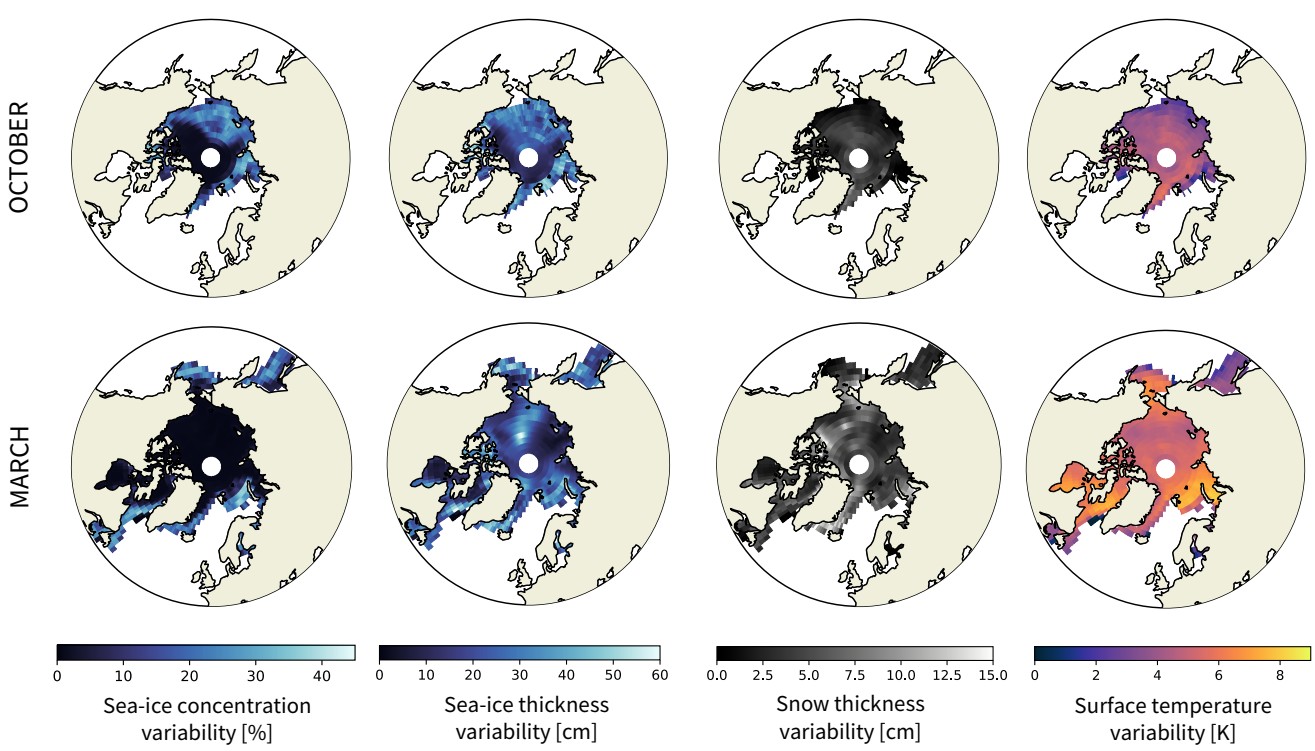

**Figure B1.** Time mean of the standard deviation for each date over a sample of five years (2003 to 2008), for each variable and each grid cell. This standard deviation was the variability estimate used to modulate the input variables in the sensitivity studies of Sec. 4.3.2. These values are inferred from the SICCI2 assimilation run.

*Author contributions.* C.B., D.N. and L.T.P. developed the original idea of this manuscript. C.B. carried out all analyses and wrote the manuscript. All authors contributed to discussions.

*Competing interests.* No competing interests are present.

5 *Acknowledgements.* We thank Stefan Kern for constructive comments and discussions. We also thank two anonymous reviewers for very constructive and insightful comments. This work was funded by the project "ESA CCI Sea Ice Phase 2".