# Peer review of "The Arctic Ocean Observation Operator for 6.9 GHz (ARC3O) - Part 2: Development and evaluation"

_The Cryosphere, 2019_

## Referee Comment (RC1) · Anonymous Referee #1 · 6 Mar 2020

Review of manuscript "The Arctic Ocean Observation Operator for 6.9 GHz (ARC3O) - Part 1: development and evaluation"

The manuscript assesses sources of uncertainty of brightness temperature from 6.9 GHz observations at top of the atmosphere. The brightness temperature was simulated using a scheme developed for this purpose, called the Arctic Ocean Observation Operator (ARC3O), which is described in details in the manuscript. This is also called observation operator as appears in the title. It comprises an earth system model with its atmospheric and oceanic components.

Results on the difference between the simulated and observed brightness tempera-

ture are presented with detailed study of the factors that contribute to the differences, including the source of the assimilated ice concentrations (three sources are tested). The study presents results in three sections to address ice conditions during cold winter, onset of melt in the spring and melting stage in summer. The contribution of the ocean and atmosphere parametrization are presented in a separate section.

The manuscript presents good and timely piece of work. Some results are very much needed in order to proceed with more accurate ice monitoring and modelling. An example is the effect of melt pond on brightness temperature. Also, the finding that variability in ice concentration estimates is the main driver for brightness temperature. Such findings set priorities for further research by both modeling and parameter retrieval communities.

I find the methodology well-planned; the manuscript is well-organized and written, graphs are well prepared and the conclusions are clear (though they can be grouped and summarized better in the Conclusions sections). This is the first study (as far as I know) that uses this simulation approach to assess the uncertainty of the microwave radiometric observations. I recommend publications after a revision that addresses the following concerns.

Comments to be considered by the authors:

In the Abstract . . . you do not really "evaluate" ARC3O. I see that you use this tool to evaluate the uncertainty of the brightness temperature and relate it to the contributing factors from ocean/ice and atmosphere. If this is true please re-phrase line 4 in the Abstract.

P 1 L7-8 "We find that they differ up to 10 K in the period between October and June, depending on the region and the assimilation run". I understand that the 3 runs differ by up to 10K. But what are differences with AMSR-E observations?

P2 L2: "by the physical noise at the level of the satellite". What is physical noise? Do

you mean electronic noise?

P2 L4: "relevant climate variables". Do you mean physical variables? I think the list of "climate" information in the next 2 lines are physical parameters of the snow-covered ice.

P2 L19: "Additionally, the climate system as a whole can be evaluated with this approach and not only individual variables". Please clarify.

P2 L24: the promise made in the statement "While we focus on the frequency of 6.9 GHz in this study, the framework proposed here ... can be extended to investigate the simulation of brightness temperatures at other frequencies ..." is offered without a substantial argument. Knowing the complexities of the microwave/snow/ice interaction at frequencies higher than 19 GHz, I would be in doubt about this promise.

P3 L20: "theoretical satellite"?? why "theoretical"?

P3 Eq. 1: the use of this equation should be declared here. I am not sure how and where this equation is used. Don't you use MEMLS to calculate surface brightness temperature?

P4 the flow chart of Fig. 1 is well-presented. But does the RTM need "bulk" snow temperature? This is not mentioned in the box of GCM. Also, what is the water vapor in this box? Atmospheric?

P6 Equations 2 and 3: please mention the basis of these equations ... empirical? Then based on what data? Or perhaps from a physical model?

P6 L13: In equation 4 and the definitions of its terms: it is strange to find the term of brine salinity in the equation of the density of seawater (inserted in line 15). Also, the brine volume fraction is defined in terms of "S" but "S" is not defined as the salinity of the ice layer. The definition of brine volume fraction is not convincing. Did you switch the numerator and denominator?

P7 L10: "we assume that the melting snow emissivity is 1...". Not sure that this assumption is reasonable. The microwave emissivity should be significantly lower than 1. Since the work has already been done using this assumption, the authors may include one line to justify or comment on this assumption.

P7 L11: the use of Eq. 1 is mentioned here for the first time. Please clarify in terms of the use of MEMLS.

P8 Fig.2: This an interesting data set that shows a clear trend although there is gap in data between the pond fraction 0.15 and 0.25.

P8 L1: "Therefore we need the brightness temperature at the ice surface . . . in summer . . .". But don't you need in other seasons as well? Or you assume zero water vapour in other seasons? Please justify the statement. But the methodology described in the rest of this paragraph is good.

P8 Section 3.2: "Ocean is not covered by 100% of sea ice". Yes, but should mention that this is a more serious problem particularly with the coarse-resolution 6.9 GHz.

P9 L2: is there a name for the "Wentz and Meissner (2000)"? is it an original model of modified from a previous model?

P9 L19: "but some characteristic features inherent to the mean model state might remain . . ." such as what? Also, can you comment on why the uncertainty of the observed brightness temperature itself is considered to be small? Nothing is mentioned in section 4.1.

P11 L11: "In order to allow for a realistic relation between ice concentration and thickness, . . .". I don't see an easy way to do this. In order to save reader's time on checking the given reference please describe in one line how this was done.

P11 L25-29: The simulated Tb are slightly higher in regions of high ice concentration and thickness, and vice versa. How high and how low? Also, would you suggest reasons to explain this observation, especially when it is coming from the 3 runs?

One would expect the difference to be small in winter season when the concentration approaches 100%.

P13 L3: "NASA Team brightness temperatures..." You mean brightness temperature from using NASA Team. Of course, NT does not produce Tb.

P13 L17: you use only the SICCI2 run to examine the sensitivity and justify the use of this single run, based on the fact that "physical relationships linking the different variables are the same in all three assimilation runs". But would the different conceptual framework in different retrieval methods play a role here?

P13 L24-28: any suggested threshold on the ice concentration that causes switching the sensitivity from the concentration to the surface temperature? Do you think this is also linked with the ice type?

P14 Table 1: this is an important contribution

P14 L8: "data assimilation on sea ice concentration ..." Is it "on" or should be "of"?

P14 L10: the difference of concentration in the MIZ can be as large as, say, 30% (not 5%)

P15 last paragraph: but cannot you evaluate the difference for cases of 100% ice concentration only? That would still be useful.

P16 L2: the difference between the Bootstrap and NT algorithms varies depending on the ice cover and season. I am not sure that Bootstrap always give higher range. You quoted 2 references. Have you checked more sources?

P17 L6: I think 2 m ice thickness is reasonable assumption. 4 m is too much. Please confirm this 4 m by quoting a reference.

P17 last paragraph: the assumption of a cell having one ice types (MY ice if ice keep circulating for more than a year) is difficult to accept. You hardly find ice circulating within one cell for more than a year. With the very large cell dimension from the 6.9

GHz observations, the cell is almost always heterogenous (MY, FY ice and OW) in highly dynamic regions such as the Beaufort Sea. I would suggest reconsidering this a possible source of error.

P19 L25: "As" instead of "Like" P19 L28: when include several references between brackets it is preferable to order them from old to recent) P19 L34: the sentence is not clear. Please rephrase.

---

## Referee Comment (RC2) · Anonymous Referee #2 · 18 Mar 2020

This is an excellent paper that advances the field of satellite emulation as a way of evaluating sea ice simulations in Earth System Models. There is one drawback to this study that the authors acknowledge in that they do not use a thickness distribution $g(h)$, but instead simulate the mean sea ice state in the coupled model chosen for the research. As a consequence, the observational operator is not able to determine the extent to which thin lead ice of thickness $h$ is important in representing brightness temperature relative to completely open water. Instead, using the methods presented, it is only possible to state that sea ice concentration is important, whereas satellites 'see' the gradation in ice thickness as represented in models with the state variable $g(h)$. This is a critical point that perhaps could be made in the conclusion of the paper

as a potential extension of this work in future studies. Aside from this, the paper was well written, easy to understand, and a technical advance in the field of polar simulation and model analysis.
* * *

---

## Author Response (AR1)

**Final Author Comments**

**The Arctic Ocean Observation Operator for 6.9 GHz (ARC3O) - Part 2: Development and evaluation**

Burgard, C., Notz, D., Pedersen, L.T., Tonboe, R.T.
*The Cryosphere,* #10.5194/tc-2019-318
* * *
**RC: Reviewer Comment**,    AR: Author Response,    *changed manuscript text*

AR: We thank both reviewers for taking the time to read through our paper with such detailed attention. We acknowledge that there was a need for precision in some parts. We are very grateful for the constructive discussion and suggestions and have made our best to fulfill your expectations and clarify your concerns. Thank you!

**1. Reviewer #1**

RC: **Reviewer Summary:**
**The manuscript assesses sources of uncertainty of brightness temperature from 6.9 GHz observations at top of the atmosphere. The brightness temperature was simulated using a scheme developed for this purpose, called the Arctic Ocean Observation Operator (ARC3O), which is described in details in the manuscript. This is also called observation operator as appears in the title. It comprises an earth system model with its atmospheric and oceanic components.**
**Results on the difference between the simulated and observed brightness temperature are presented with detailed study of the factors that contribute to the differences, including the source of the assimilated ice concentrations (three sources are tested). The study presents results in three sections to address ice conditions during cold winter, onset of melt in the spring and melting stage in summer. The contribution of the ocean and atmosphere parametrization are presented in a separate section.**
**The manuscript presents good and timely piece of work. Some results are very much needed in order to proceed with more accurate ice monitoring and modelling. An example is the effect of melt pond on brightness temperature. Also, the finding that variability in ice concentration estimates is the main driver for brightness temperature. Such findings set priorities for further research by both modeling and**

**parameter retrieval communities.**

**I find the methodology well-planned; the manuscript is well-organized and written, graphs are well prepared and the conclusions are clear (though they can be grouped and summarized better in the Conclusions sections). This is the first study (as far as I know) that uses this simulation approach to assess the uncertainty of the microwave radiometric observations. I recommend publications after a revision that addresses the following concerns.**

AR: Thank you very much for the positive feedback, and for your detailed, constructive comments on how to further improve our manuscript.
We have rearranged a little the Conclusions section by dividing it in a "Conclusions" and an "Outlook" section. Additionally, we added one paragraph summarizing the workflow of ARC3O.
Also, we have addressed all your other comments as described in the following.

RC: **In the Abstract... you do not really "evaluate" ARC3O. I see that you use this tool to evaluate the uncertainty of the brightness temperature and relate it to the contributing factors from ocean/ice and atmosphere. If this is true please re-phrase line 4 in the Abstract.**

AR: Our setup does indeed not allow us to evaluate ARC3O directly. However, through understanding the uncertainty of the brightness temperature and its drivers, we indirectly evaluate the results of ARC3O. We have reformulated as follows:
*To evaluate sources of uncertainties when applying ARC3O, we compare brightness temperatures simulated by applying ARC3O on three assimilation runs of the MPI Earth System Model (MPI-ESM), assimilated with three different sea-ice concentration products, with brightness temperatures measured by the Advanced Microwave Scanning Radiometer Earth Observing System (AMSR-E) from space.*

RC: **P1 L7-8 "We find that they differ up to 10 K in the period between October and June, depending on the region and the assimilation run". I understand that the 3 runs differ by up to 10K. But what are differences with AMSR-E observations?**

AR: We apologize for the confusion. We mean that the individual runs differ by up to 10 K from AMSR-E observations. We have reformulated as follows:
*We find that the simulated and observed brightness temperatures differ up to 10 K in the period between October and June, depending on the region and the assimilation run.*

RC: **P2 L2: "by the physical noise at the level of the satellite". What is physical noise? Do you mean electronic noise?**

AR:   Yes, we mean electronic noise and now say so in the manuscript.

**RC:   P2 L4: "relevant climate variables". Do you mean physical variables? I think the list of "climate" information in the next 2 lines are physical parameters of the snow-covered ice.**

AR:   Thank you for pointing out this imprecision. It is true that we only describe physical properties of the snow-covered ice (having in mind that these are the important ones for 6.9 GHz). However, for higher frequencies, atmospheric information becomes important as well. This is what we hinted at. We have reformulated as follows:
*The contribution of the individual drivers on the brightness temperature cannot be disentangled unambiguously if the properties describing the state of the ice, snow, open ocean surface, and atmosphere are not available simultaneously.*

**RC:   P2 L19: "Additionally, the climate system as a whole can be evaluated with this approach and not only individual variables". Please clarify.**

AR:   Here, we mean that the translation from model state into brightness temperature requires more than one physical parameter. This way, the evaluation of the simulated brightness temperature is in fact an evaluation of the physical climate state as given by the combination of several variables instead of an evaluation of an individual variable taken out of the climatic context. We reformulated as follows:
*Additionally, the simulation of the brightness temperature relies on several physical variables. Therefore, the evaluation of the simulated brightness temperature is an evaluation of the physical climate state as a combination of several variables, allowing an integrated evaluation of the simulated climate state of the GCM instead of a "variable-to-variable" evaluation*

**RC:   P2 L24: the promise made in the statement "While we focus on the frequency of 6.9GHz in this study, the framework proposed here...can be extended to investigate the simulation of brightness temperatures at other frequencies..." is offered without a substantial argument. Knowing the complexities of the microwave/snow/ice interaction at frequencies higher than 19 GHz, I would be in doubt about this promise.**

AR:   We agree that the exact same methodology cannot be used as easily for higher frequencies due to the challenges of representing snow accurately. However, we believe that further research and revisiting old and new in-situ measurements can help us to simplify some of the governing snow properties. And, using an idealised setup similar to the setup we used in the companion manuscript of this manuscript, we might find a simplified way of describing snow properties relevant to the simulation of brightness temperatures. Again, we agree, this could not be done within the next few months. But maybe in the next years

to decades? We have added the following disclaimer:
*However, the increasing influence of snow on the brightness temperature with increasing frequency and the limited possibilities of simulating snow properties in a GCM remain a challenge to overcome first.*

**RC:  P3 L20: "theoretical satellite"?? why "theoretical"?**

 AR:  We chose to write "theoretical" because a satellite will never be able to rotate around a climate model, which is a theoretical construction. We realize that the information about the climate model is missing in this sentence. We have clarified that our observation operator is defined to be applied to climate model output, which justifies the use of "theoretically":
*The purpose of the ARCtic Ocean Observation Operator for 6.9 GHz (ARC3O) is to simulate Arctic Ocean brightness temperatures as could be measured theoretically at the top of the atmosphere of a climate model by a satellite rotating around that climate model.*

**RC:  P3 Eq. 1: the use of this equation should be declared here. I am not sure how and where this equation is used. Don't you use MEMLS to calculate surface brightness temperature?**

 AR:  We apologize for the confusion. We have moved the equation to Sec. 3.1.3 (Melting snow), where we use this equation to clarify the relationship between emissivity and brightness temperature and to compute the brightness temperature of ice covered by melting snow.

**RC:  P4 the flow chart of Fig.1 is well-presented. But does the RTM need "bulk" snow temperature? This is not mentioned in the box of GCM. Also, what is the water vapor in this box? Atmospheric?**

 AR:  The atmospheric RTM does not need the bulk snow temperature, but rather the surface temperature of the snow-ice column. Combined with the ocean surface temperature, this snow-ice column surface temperature gives us an approximation of the mean air temperature near the surface. All other information about the snow-ice column is already comprised in the ice brightness temperature computed by MEMLS. The water vapor is atmospheric. We have clarified this in the last box in Fig. 1.

**RC:  P6 Equations 2 and 3: please mention the basis of these equations...empirical? Then based on what data? Or perhaps from a physical model?**

 AR:  These equations are a fit based on the results of the physical model SAMSIM (Griewank and Notz, 2015), which describes the evolution of salinity in sea ice in a 1D setup.

The model results were compared to observational data for evaluation. We used these equations for the "salinity as a function of depth" in our companion manuscript. We have added the following few words about the origin of these equations in the manuscript:

*The salinity profile $S$ is computed as a function of depth $z$, as formulated by Griewank and Notz (2015), based on the results of 1D simulations with the complex thermodynamic sea-ice model SAMSIM and their comparison to observations.*

**RC:** **P6 L13: In equation 4 and the definitions of its terms: it is strange to find the term of brine salinity in the equation of the density of seawater (inserted in line 15). Also, the brine volume fraction is defined in terms of "S" but "S" is not defined as the salinity of the ice layer. The definition of brine volume fraction is not convincing. Did you switch the numerator and denominator?**

**AR:** We apologize for the confusion. The brine liquid is defined as "seawater" here because it is a liquid with a similar chemical composition as seawater (since it is a result of the freezing process of seawater). And the salinity of this liquid is brine salinity. This is why we use brine salinity in that equation.

S is the salinity of the ice layer, depending on the depth (see Eq. 2 and 3). The brine volume fraction is defined following the equation given in Notz (2005), we did not switch numerator and denominator. To avoid confusion, we have restructured this part, adding all steps of the computation. Also, we corrected a mistake in the formula we had before in the manuscript for brine salinity.

*See Page 6 of the manuscript.*

**RC:** **P7 L10: "we assume that the melting snow emissivity is 1...". Not sure that this assumption is reasonable. The microwave emissivity should be significantly lower than 1. Since the work has already been done using this assumption, the authors may include one line to justify or comment on this assumption.**

**AR:** At 6.9 GHz, the emissivity of wet snow is very high (Hallikainen et al. 1986, IEEE on Antennas and Propagation, Lee et al. 2018, JGR: Atmospheres). Experiments conducted by one of our co-authors also show an emissivity of wet snow near 1. We have included the two references mentioned above in the manuscript:

*At 6.9 GHz, the emitting part of wet snow is a thin subsurface layer and the emissivity is close to 1 (Hallikainen et al. 1986; Lee et al. 2018).*

**RC:** **P7 L11: the use of Eq. 1 is mentioned here for the first time. Please clarify in terms of the use of MEMLS.**

**AR:** We use Eq.1 here because in the case of wet snow we cannot use MEMLS due to a lack of information about the liquid water content in the snow. Instead, we rely on the

relationship between temperature and emissivity, setting the emissivity to 1. As explained in a previous answer, we have defined and clarified the use of Eq.1 (now Eq. 8) in this part. It now reads:

*However, these snow properties are not resolved in MPI-ESM. We therefore cannot use MEMLS to simulate the brightness temperature of the ice and snow column in this case. Instead, we use the following definition of the brightness temperature:*

$$TB = \epsilon_{eff} \cdot T_{eff} \tag{1}$$

*where $\epsilon_{eff}$ is the emissivity of the emitting part of the ice and snow column, i.e. the layers influencing the resulting radiation emitted at the surface, and $T_{eff}$ the integrated temperature over this same emitting part (Hallikainen and Winebrenner, 1992; Tonboe, 2010; Shokr and Sinha, 2015). At 6.9 GHz, the emitting part of wet snow is a thin subsurface layer and the emissivity is close to 1 (Hallikainen et al. 1986; Lee et al. 2018). Following Eq. 8, we therefore assume that the brightness temperature of ice covered by melting snow is equal to the snow surface temperature.*

**RC: P8 Fig.2: This an interesting data set that shows a clear trend although there is gap in data between the pond fraction 0.15 and 0.25**

AR: We agree. Unfortunately, we do not have the data to fill the gap.

**RC: P8 L1: "Therefore we need the brightness temperature at the ice surface...in summer...". But don't you need in other seasons as well? Or you assume zero water vapour in other seasons? Please justify the statement. But the methodology described in the rest of this paragraph is good.**

AR: We apologize for the confusion and recognize the need for further clarification here. In other seasons, MEMLS computes the ice brightness temperature, which is then used directly in the atmospheric RTM, giving us the brightness temperature at the top of the atmosphere.

For the summer ice brightness temperature, we use empirical data, which is measured at the top of the atmosphere. We therefore cannot feed this to the atmospheric RTM, because the atmosphere would be taken into account twice in this case. Instead, we want to come back to a value representing the ice surface brightness temperature that can be combined with the ocean brightness temperature, and then the atmospheric effect, in the RTM. This is why we apply this procedure in summer only. We have rethought the structure of this explanation and think it might be more logical to include this in the Section about the atmospheric radiative transfer model. We therefore only briefly point to it here as follows:

*After applying an atmospheric correction (see Sec. 3.2), the resulting summer ice brightness temperature at the surface is 266.78 K.*

And then reformulated as follows in Sec. 3.2:

*To infer a mean atmospheric correction, we apply the geophysical model to regions covered by 99% or more sea ice in MPI-ESM output presented in Sec.4.2, setting all melt pond fractions to zero. This way, we have no influence by open water surfaces, be it ocean or melt ponds, on the resulting brightness temperature. We set the ice surface brightness temperature used as input for the geophysical model to a random constant. We then subtract this constant ice surface brightness temperature from the top-of-the-atmosphere brightness temperature simulated by the geophysical model based on atmospheric properties given by the climate model output. This gives us a mean atmospheric effect of 4.49 K. We add this to the brightness temperature of 262.29 K inferred in Sec.3.1.4, resulting in a constant brightness temperature of 266.78 K as a constant brightness temperature representing the radiation emitted at the summer bare ice surface. This is the bare ice summer surface brightness temperature that can be used for combination with open water (ocean and melt ponds) brightness temperature in the geophysical model in Step 5 of Fig.1.*

**RC:** **P8 Section 3.2: "Ocean is not covered by 100% of sea ice". Yes, but should mention that this is a more serious problem particularly with the coarse-resolution 6.9 GHz.**

AR: The resolution of the satellite footprint is not an issue for the simulation of the brightness temperature of a mixed ice-ocean surface. It is rather the difference between model and satellite resolution in the later comparison that is a challenge. Still, we have reformulated as follows to highlight the importance of the heterogeneity of the surface for the brightness temperature simulation:

*As the Arctic Ocean is not covered by 100% of sea ice, the brightness temperature measured at the top of the atmosphere is also influenced by the relative fraction and properties of open water surfaces and properties of the atmosphere. To take into account these oceanic and atmospheric contributions, we use a geophysical model developed by Wentz and Meissner (2000) (Step 5 in Fig. 1).*

**RC:** **P9 L2: is there a name for the "Wentz and Meissner (2000)"? is it an original model or modified from a previous model?**

AR: It is an original model developed to support their "Ocean retrieval algorithm" for AMSR measurements. There is no specific name for this model as far as we know of.

**RC:** **P9 L19: "but some characteristic features inherent to the mean model state might remain..." such as what? Also, can you comment on why the uncertainty of the observed brightness temperature itself is considered to be small? Nothing is mentioned in section 4.1.**

AR: By "characteristic features inherent to the mean model state", we mean that the mean model state and the assimilated state are so incompatible that the model will drift towards mean model state if the assimilation step is too long. Let's say, for example, that the model has a warm bias in a given region. If a non-zero sea-ice concentration is assimilated into that region, the sea ice might melt completely until the next step because the ocean is too warm at that place for sea ice to exist. Such effects are minimized by assimilating several climate variables at the same time. However, more complex relationships might be at work which cannot be compensated by assimilating most of the variables from reanalysis.
We have added the following sentence for clarification:
*This is the case when the mean model state and the assimilated state are so incompatible that the model will rapidly drift back towards the mean model state.*
Additionally, we have added a sentence about the uncertainty of AMSR measurements:
*The uncertainty of the measured brightness temperature itself is around 1 K (NASDA, 2003) and thus neglected here.*

RC: **P11 L11: "In order to allow for a realistic relation between ice concentration and thickness,...". I don't see an easy way to do this. In order to save reader's time on checking the given reference please describe in one line how this was done.**

AR: As mentioned, the method is very simple and the reference shows that it outperforms other methods used for data assimilation of ice thickness. We have added the following description:
*The assimilation changes the thickness $h$ in the given grid cell by $\Delta h_{assim}$, which is proportional to $\Delta SIC_{assim}$, with a proportionality factor $h*$ of 2 m, as follows:*

$$\Delta h_{assim} = h * \Delta SIC_{assim} \qquad (2)$$

RC: **P11 L25-29: The simulated Tb are slightly higher in regions of high ice concentration and thickness, and vice versa. How high and how low? Also, would you suggest reasons to explain this observation, especially when it is coming from the 3 runs? One would expect the difference to be small in winter season when the concentration approaches 100%.**

AR: We have investigated these differences from several perspectives, trying to link them with parameters used as input for ARC3O. The only explanation that can explain the pattern and magnitude for too high brightness temperatures in the regions North of Canada (by about 3 to 5 K) is the too low ice thickness in the simulation, likely induced by the assimilation process, as explained later in the manuscript.
Regarding the regions of too low brightness temperatures, they are mainly marginal zones with seasonal ice cover, reaching an underestimation of 10 to 15 K in some cases.

However, here, the too low brightness temperatures only occur when the brightness temperatures are simulated based on NASA Team or SICCI sea-ice concentrations, not when they are simulated based on the Bootstrap sea-ice concentrations. As can be seen in the last row of Fig. 8, the sea-ice concentration is much lower in these regions in the NT and SICCI product than in Bootstrap. These differences are therefore rather a result of differences in the retrieved sea-ice concentration than a problem in the simulation.

**RC:** **P13 L3: "NASA Team brightness temperatures..." You mean brightness temperature from using NASA Team. Of course, NT does not produce Tb.**

AR: Yes of course. This has been corrected.

**RC:** **P13 L17: you use only the SICCI2 run to examine the sensitivity and justify the use of this single run, based on the fact that "physical relationships linking the different variables are the same in all three assimilation runs". But would the different conceptual framework in different retrieval methods play a role here?**

AR: To confirm that the choice of assimilation run would not strongly influence our qualitative conclusion, we have run the sensitivity analysis for Bootstrap and NASA Team as well. Except very locally and one larger region in the NASA Team assimilation run, all support the prevailing importance of sea-ice concentration in regions with less than 100% sea-ice concentration and the prevailing importance of surface temperature in regions of 100% sea-ice concentration. We have reformulated as follows:
*We do so for the SICCI2, the Bootstrap, and the NASA Team assimilation run. As the results of the analysis are similar for all three assimilation runs unless otherwise mentioned, we only show the results for the SICCI2 assimilation run in the following.*
and have added the following paragraph:
*In the sensitivity study using the NASA Team assimilation run, the variability range of sea-ice thickness and surface temperature are comparable to the other runs (not shown). However, the sea-ice thickness is the main driver of variability for a large region in the Central Arctic north of Alaska in March (not shown). Further investigation into these differences in the main driver for the brightness temperature could lead to a better understanding of the differences in the simulated climate of these three assimilation runs. However, this is beyond the scope of our study and is a subject for future work.*

**RC:** **P13 L24-28: any suggested threshold on the ice concentration that causes switching the sensitivity from the concentration to the surface temperature? Do you think this is also linked with the ice type?**

AR: We think the threshold where it switches to surface temperature is for sea-ice concentrations close to 100%, with low variabilities throughout the years. This is because the sea-ice concentration has such a high influence in the other regions that the variability has to remain low to allow another parameter to have a higher influence. In numbers: Considering that water TB is around 160 K and ice TB is around 260 K, the sensitivity of the total Tb to changes in sea-ice concentration is close to 1 K per 1 % of sea-ice concentration. As the penetration depth at 6.9 GHz is high, the sensitivity of the total Tb to the surface temperature will be significantly less than 1K per 1K. We have not checked yet if this is linked to ice types but plan to look into it in future work.

**RC:  P14 Table 1: this is an important contribution**

 AR:  Thank you.

**RC:  P14 L8: "data assimilation on sea ice concentration..." Is it "on" or should be "of"?**

 AR:  It should be "on". We want to investigate the effect of the method of data assimilation on the sea-ice simulated by the model. We have added "procedure" in the sentence to clarify:
*As the sea-ice concentration is the main driver for uncertainties in the brightness temperature simulation, we here focus on the effect of the data assimilation procedure on the sea-ice concentration in the three different assimilation runs.*

**RC:  P14 L10: the difference of concentration in the MIZ can be as large as, say, 30% (not 5%)**

 AR:  We do not mean differences in sea-ice concentration between the observational products. Instead, we mean the difference between the observational product and the assimilated sea-ice concentration. With an ideal assimilation method and ideal model, the observed and the assimilated sea-ice concentration should be the same. Here, they are not, however, but the differences between the product and the assimilated ice only reach 5% in marginal regions (see Fig. 6) We have reformulated as follows:
*This effect is mostly visible in the marginal regions (Fig. 6) and is of similar magnitude for all three assimilation runs. At the ice edge, the differences between the original sea-ice concentration observational product and the sea-ice concentration assimilated in the simulation are highest, on the order of 5 %.*

**RC:  P15 last paragraph: but cannot you evaluate the difference for cases of 100% ice concentration only? That would still be useful**

 AR:  To do so, we would need to be sure that the location chosen have exactly the same properties, such as concentration, thickness, surface temperature in reality and the model at the same time, which is not straightforward.

**RC:** **P16 L2: the difference between the Bootstrap and NT algorithms varies depending on the ice cover and season. I am not sure that Bootstrap always give higher range. You quoted 2 references. Have you checked more sources?**

**AR:** We agree that we cannot guarantee that the real sea-ice concentration is always between Bootstrap and NASA Team. However, we still think that the spread between the two is a good estimate of the uncertainty in sea-ice concentration products. We have reformulated as follows:
*To do so, we use the spread between the SICCI2, Bootstrap and NASA Team sea-ice concentration as an estimate of the uncertainty range around the real sea-ice concentration (Fig.7 and Ivanova et al. (2014); Kern et al. 2019).*
and highlighted that this is an assumption:
*Finally, we treat the spread between the SICCI2, Bootstrap and NASA Team sea-ice concentration as an estimate of the uncertainty range around the real sea-ice concentration. Hence, we implicitly assumed that the real sea-ice concentration lies in the range between the SICCI2, the Bootstrap and the NASA Team estimates. As only limited evaluation against reality is possible, the real sea-ice concentration could lie outside of this range and the uncertainty between real and retrieved sea-ice concentration might be different to what we assume here.*

**RC:** **P17 L6: I think 2 m ice thickness is reasonable assumption. 4 m is too much. Please confirm this 4 m by quoting a reference**

**AR:** Our use of 4 m was to check if the ice thickness can have such an impact on the brightness temperature. The assumption of around 4 m thickness of the thickest ice north of Greenland and the Canadian Archipelago is based on the new SMOS/CryoSat2 product (see Fig.9 in Ricker et al., 2017, The Cryosphere or the quicklook: https://spaces.awi.de/pages/viewpage.action?pageId=291898639). The product shows rather a thickness of between 3 and 4 m. We have reformulated as follows:
*Observational estimates tend to show thicknesses of rather 3 to 4 m in this region (Ricker et al., 2017).*

**RC:** **P17 last paragraph: the assumption of a cell having one ice types (MY ice if ice keep circulating for more than a year) is difficult to accept. You hardly find ice circulating within one cell for more than a year. With the very large cell dimension from the 6.9 GHz observations, the cell is almost always heterogeneous (MY, FY ice and OW) in highly dynamic regions such as the Beaufort Sea. I would suggest reconsidering this a possible source of error.**

**AR:** We agree and have highlighted this issue further as follows:

*Rethinking this definition, e.g. by finding a way to follow the movement of the ice in the climate model, might therefore also reduce the uncertainty.*

**RC:** **P19 L25: "As" instead of "Like"**

AR: Changed.

**RC:** **P19 L28: when include several references between brackets it is preferable to order them from old to recent)**

AR: Noted, we have checked the manuscript for such occurrences.

**RC:** **P19 L34: the sentence is not clear. Please rephrase.**

AR: Noted, we have reformulated as follows:
*The constant bare ice brightness temperature of 266.78 K is derived by using observed brightness temperatures and melt-pond fractions, as explained in Sec. 3.1.4).*

**2. Reviewer #2**

RC: **Reviewer summary:**
**This is an excellent paper that advances the field of satellite emulation as a way of evaluating sea ice simulations in Earth System Models. There is one drawback to this study that the authors acknowledge in that they do not use a thickness distribution g(h), but instead simulate the mean sea ice state in the coupled model chosen for the research. As a consequence, the observational operator is not able to determine the extent to which thin lead ice of thickness his important in representing brightness temperature relative to completely open water. Instead, using the methods presented,it is only possible to state that sea ice concentration is important, whereas satellites 'see' the gradation in ice thickness as represented in models with the state variable g(h). This is a critical point that perhaps could be made in the conclusion of the paper as a potential extension of this work in future studies. Aside from this, the paper was well written, easy to understand, and a technical advance in the field of polar simulation and model analysis.**

AR: Thank you very much for the positive feedback. We have now included the matter of the ice thickness distribution in the conclusion, as follows:

[revised manuscript text omitted]